# Identifying Learnwares via Reduced Neural Conditional Mean Embedding

**Zi-Yu Mao** [1 2]  **Ming Li** [1 2]

## Abstract

The learnware paradigm aims to establish a market of learnwares, each of which is a well-trained model combined with a specification that describes its functionality without leaking data privacy. The market enables users to efficiently reuse relevant models based on specifications on their own tasks instead of training models from scratch. Recent works have attempted to generate specifications using Reduced Kernel Mean Embedding (RKME), which maps input distributions into Reproducing Kernel Hilbert Space (RKHS) while ignoring the output space, causing models trained on similar input spaces to yield similar specifications, even when their functionalities differ. Many labeled-RKME improvements attempt to address this by indirectly modeling the input-output conditional distributions, but they remain limited to classification tasks and lack clear theoretical explanations. In this work, we propose Reduced Neural Conditional Mean Embedding (RNCME), a novel specification generation method that directly models input-output conditional distributions via Conditional Mean Embedding (CME). Our RNCME method has a clear theoretical understanding based on CME and is applicable to both regression and classification tasks. Empirical experiments demonstrate the effectiveness and efficiency of our RNCME method.

## 1. Introduction

"Learnware = Model + Specification" (Zhou, 2016; Zhou & Tan, 2024). In the learnware paradigm, each learnware consists of a well-trained machine learning model accompanied by a specification that describes its functionality. This shared learnware market enables users to efficiently identify and reuse relevant learnwares based on specifications, eliminating the need for expensive model training from scratch, which requires abundant training data, substantial computational resources, and specialized expertise. It is worth noting that the learnware market has no access to either the data of developers or users: developers only need to submit well-trained models and their specifications, while users are required to provide only task specifications and will receive suitable learnware(s) from the market based on specifications, addressing privacy concerns in traditional model reuse approaches.

It is obvious that the specification plays a crucial role in the learnware paradigm. Wu et al. (2023) proposed to generate the specification based on the Reduced Kernel Mean Embedding (RKME), which maps input distributions into a Reproducing Kernel Hilbert Space (RKHS). Specifically, for a model $f$ well-trained on a dataset $(X, Y)$, RKME first estimates the empirical Kernel Mean Embedding (KME) for the marginal distribution $P_X$ on dataset $X$, and then generates a reduced set as the specification, which approximates the empirical KME. Moreover, Lei et al. (2024) provided a theoretical analysis of RKME specification about its preservation ability for training data, ensuring data privacy preservation in RKME specifications.

The basic RKME method focuses solely on the input space and neglects the output space, causing models trained on similar input spaces receive similar specifications even if their functionalities differ, and vice versa. For example, consider a user with a "Cartoon Character Emotion Prediction" task, and there are two candidate learnwares: $f_1$ is trained on similar cartoon faces but for character recognition, and $f_2$ is trained on human faces but for the same emotion prediction. If we rely solely on RKME that only considers the input distribution, the market would identify $f_1$ as the best match because its input features align perfectly with the user's data. However, $f_2$ is actually more suitable for the user's objective; despite the "style" of the input being different, the underlying logic of identifying expressions (e.g., the curve of a mouth or the narrowing of eyes) is shared between the user's task and $f_2$. This suggests that a specification ignoring the output space may prioritize distributional similarity over functional utility, leading to sub-optimal model reuse. Several recent improvements to it have incorporated output space information: Tan et al.

[1]State Key Laboratory for Novel Software Technology, Nanjing University, Nanjing, China [2]School of Artificial Intelligence, Nanjing University, Nanjing , China. Correspondence to: Ming Li <lim@lamda.nju.edu.cn>.

*Proceedings of the 43rd International Conference on Machine Learning*, Seoul, South Korea. PMLR 306, 2026. Copyright 2026 by the author(s).

(2024a) evolve heterogeneous feature spaces into a unified subspace with less entangled class representations and more coherent embeddings; Liu et al. (2025) generate class-wise RKME specification and model the utility of a learnware combination as a minimum-cost maximum-flow problem; and Guo et al. (2023) distill models into linear proxies and recommend models via bipartite graph matching. However, these methods lack clear theoretical explanations and are inherently constrained to classification tasks.

In this work, we propose a novel specification generation method called Reduced Neural Conditional Mean Embedding (RNCME), which directly models the input-output conditional distribution through Conditional Mean Embedding (CME) (Song et al., 2009). Unlike previously mentioned labeled-RKME improvements, RNCME has a clear theoretical framework rooted in CME theory and handles both regression and classification tasks. To address the well-known computational bottleneck of standard CME, we employ a finite-dimensional kernel for the input space. While a finite-dimensional RKHS generally lacks universal approximation capability and characteristic properties, we overcome this limitation by adopting a data-adaptive neural feature map, which is optimized in a specialized training phase to capture essential input-output mapping patterns. Moreover, our neural feature map eliminates the need for manual kernel selection, including the choice of kernel functions and the tuning of regularization parameters. To preserve data privacy, we propose an iterative algorithm that solves a constrained vector-weighted low-rank kernel approximation problem. This algorithm generates a reduced set that closely approximates the empirical CME. We further establish a theoretical upper bound on the associated approximation error, providing guarantees on the fidelity of the reduction process. Experiments on both synthetic and real-world datasets, including comparative evaluations and ablation studies, demonstrate the effectiveness and efficiency of RNCME. We summarize the main contributions of our work as follows:

- We propose RNCME for specification generation, which directly models input-output conditional distributions with a clear theoretical understanding based on CME, extending existing labeled-RKME improvements to both regression and classification;

- We design a data-adaptive neural feature map, which reduces the high computational complexity of empirical CME estimation while avoiding the manual kernel selection problem;

- We propose an algorithm for solving the constrained vector-weighted low-rank kernel approximation problem, which is used to generate a privacy-preserving reduced set that approximates the empirical CME.

The remainder of this paper is structured as follows. Section 2 provides essential background and preliminaries. Section 3 introduces the proposed RNCME method. The experimental evaluation and results are detailed in Section 4, while Section 5 discusses connections to prior research. Finally, concluding remarks are presented in Section 6.

## 2. Preliminaries

In this section, we briefly introduce some preliminaries, beginning with the definition of notations.

### 2.1. Notations

Let $X$ be a random variable taking values in a measurable space $\mathcal{X}$, with realizations $x$. Consider a measurable positive definite kernel $k_{\mathcal{X}} : \mathcal{X} \times \mathcal{X} \to \mathbb{R}$ that induces a Reproducing Kernel Hilbert Space (RKHS) $\mathcal{H}$, endowed with inner product $\langle \cdot, \cdot \rangle_{\mathcal{H}}$ and norm $\|\cdot\|_{\mathcal{H}}$. The canonical feature map $\phi : \mathcal{X} \to \mathcal{H}$ satisfies $\phi(x) = k_{\mathcal{X}}(x, \cdot)$ for all $x \in \mathcal{X}$. And the reproducing property of $k_{\mathcal{X}}$ yields $h(x) = \langle h, \phi(x) \rangle_{\mathcal{H}}$ holds for all $h \in \mathcal{H}$ and $x \in \mathcal{X}$.

Analogously, for random variable $Y$ in measurable space $\mathcal{Y}$ with kernel $k_{\mathcal{Y}}$, we denote its induced RKHS as $\mathcal{G}$ with feature map $\psi : \mathcal{Y} \to \mathcal{G}$, inner product $\langle \cdot, \cdot \rangle_{\mathcal{G}}$, and norm $\|\cdot\|_{\mathcal{G}}$, satisfying the corresponding reproducing property.

### 2.2. The Learnware Paradigm

The learnware paradigm can be divided into two stages: submitting and deploying stages.

**The Submitting Stage** In this stage, all developers can submit their well-performing models to the learnware market spontaneously. Suppose there are $N$ developers, and the $c$-th developer is accessible to a local dataset $D_c = (X_c, Y_c) = \{(x_{ci}, y_{ci})\}_{i=1}^{n_c}$ sampled i.i.d. from the joint distribution $P_{XY}^c$ over $\mathcal{X}_c \times \mathcal{Y}_c$, where $\mathcal{X}_c \subseteq \mathcal{X}$ and $\mathcal{Y}_c \subseteq \mathcal{Y}$. Based on $D_c$, the developer can locally train a well-performing model $f_c : \mathcal{X}_c \to \mathcal{Y}_c$ and generate its corresponding specification $\mathcal{S}_c$. Then, the developer can provide $f_c$ along with its specification $\mathcal{S}_c$ to the learnware market. With $N$ such submissions, the learnware market becomes $\mathcal{M} = \{(f_c, \mathcal{S}_c)\}_{c=1}^{N}$.

**The Deploying Stage** When a user wants to solve a new task $t$ with a small labeled dataset $D_l = (X_l, Y_l) = \{(x_{li}, y_{li})\}_{i=1}^{n_l}$ and a large unlabeled dataset $D_u = X_u = \{x_{ui}\}_{i=1}^{n_u}$, they can locally generate a specification $\mathcal{S}_t$ that describes the requirements of the task based on $D_l$ (or incorporating $D_u$). The user then submits $\mathcal{S}_t$ to the market. Upon receiving the specification, the market identifies relevant learnwares based on the specifications, and recommends the most suitable learnware(s) for the user to solve task $t$.

## 2.3. Kernel Mean Embedding

In the absence of ambiguity, we use $\mathbb{E}_X, \mathbb{E}_{XY}$ instead of $\mathbb{E}_{X \sim P_X}, \mathbb{E}_{(X,Y) \sim P_{XY}}$, and so on. For the marginal distribution $P_X$, the KME (Smola et al., 2007) is defined as:

$$\mu_X = \mathbb{E}_X[\phi(X)] \in \mathcal{H}.$$

If kernel $k_{\mathcal{X}}$ is characteristic, then $\mu_X$ is unique for $P_X$. For a dataset $\{x_i\}_{i=1}^n$ sampled i.i.d. from $P_X$, the $\mu_X$ can be estimated empirically as:

$$\widehat{\mu}_X = \frac{1}{n} \sum_{i=1}^n \phi(x_i).$$

The KME of the joint distribution $P_{XY}$, also known as the cross-covariance operator (Baker, 1973), is defined as:

$$C_{XY} = \mathbb{E}_{XY}[\psi(Y) \otimes \phi(X)] \in \mathcal{G} \otimes \mathcal{H},$$

where $\otimes$ is the tensor product operator, and space $\mathcal{G} \otimes \mathcal{H}$ is isomorphic to the Hilbert-Schmidt space $\mathrm{HS}(\mathcal{H}, \mathcal{G})$.

## 2.4. Reduced Kernel Mean Embedding

KME is a potential specification for the learnware market, but it depends on the raw data violates the data privacy requirement. To address this issue, Wu et al. (2023) proposed the RKME by using a reduced set $(\boldsymbol{\beta}, \boldsymbol{Z}) = \{(\beta_j, z_j)\}_{j=1}^m$ to approximate the empirical KME of the raw data, where $\beta_j \in \mathbb{R}$ and $z_j \in \mathcal{X}$, which can be generated by the following optimization problem:

$$\min_{\boldsymbol{\beta}, \boldsymbol{Z}} \left\| \frac{1}{n} \sum_{i=1}^n \phi(x_i) - \sum_{j=1}^m \beta_j \phi(z_j) \right\|_{\mathcal{H}}^2.$$

The reduced set $(\boldsymbol{\beta}, \boldsymbol{Z})$ will be submitted to the market, and the corresponding RKME $\widetilde{\mu} = \sum_{j=1}^m \beta_j \phi(z_j)$ typically serves as the specification. RKME only considers the input space while ignoring the output space. A possible solution is to use Condition Mean Embedding (CME).

## 2.5. Conditional Mean Embedding

The KME of the conditional distributions $P_{Y|X}$, usually called Conditional Mean Embeddings (CME) (Song et al., 2009), is defined as:

$$\mathcal{U}_{Y|X} = \mathbb{E}_{Y|X}[\psi(Y)] = C_{YX} C_{XX}^{-1},$$

which is an operator from $\mathcal{H}$ to $\mathcal{G}$. For a fixed $x \in \mathcal{X}$, the CME of $P_{Y|X=x}$ is defined as:

$$\mathcal{U}_{Y|X=x} = \mathbb{E}_{Y|X=x}[\psi(Y)] = \mathcal{U}_{Y|X} \phi(x) \in \mathcal{G}.$$

For an i.i.d. dataset $\{(x_i, y_i)\}_{i=1}^n \sim P_{XY}$, define the implicit feature matrices $\Phi = \begin{pmatrix} \phi(x_1) & \phi(x_2) & \cdots & \phi(x_n) \end{pmatrix}$,

$\Psi = \begin{pmatrix} \psi(y_1) & \psi(y_2) & \cdots & \psi(y_n) \end{pmatrix}$ and the Gram matrix $K_X = \Phi^{\mathrm{T}} \Phi \in \mathbb{R}^{n \times n}$ with entries $(K_X)_{ij} = k_{\mathcal{X}}(x_i, x_j)$; given the regularization parameter $\lambda > 0$ and the $n \times n$ identity matrix $I$, the empirical CME $\widehat{\mathcal{U}}_{Y|X}$ (simply denoted as $\widehat{\mathcal{U}}$) can be estimated as:

$$\widehat{\mathcal{U}} = \Psi \left( K_X + n\lambda I \right)^{-1} \Phi^{\mathrm{T}}. \tag{1}$$

# 3. Reduced Neural CME

We propose to directly model input-output conditional distributions via CME, which has a clear theoretical understanding, and handles both regression and classification tasks.

## 3.1. Address High Computational Complexity

Theoretically, the empirical CME can be taken in the form:

$$\widehat{\mathcal{U}} = \widehat{C}_{YX} \left( \widehat{C}_{XX} + \lambda \mathcal{I} \right)^{-1}, \tag{2}$$

where $\mathcal{I}$ is the identity operator; $\widehat{C}_{YX} = \frac{1}{n} \sum_{i=1}^n \psi(y_i) \otimes \phi(x_i)$ and $\widehat{C}_{XX}$ admits a similar expansion.

The kernel trick avoids the need for explicit feature maps by using a kernel function to compute inner products, which makes it possible to use infinite-dimensional kernels with universal and characteristic properties (e.g., RBF kernel), and rewrites $\widehat{\mathcal{U}}$ as Equation (1) (see Appendix A for proof).

However, inverting the $n \times n$ matrix requires $O(n^3)$ computational complexity, which is infeasible for large $n$. Therefore, we propose a pragmatic compromise by using an explicitly finite-dimensional version $\phi_d : \mathcal{X} \to \mathbb{R}^d$, where $d \ll n$, and the corresponding kernel is defined as $k_d(x, x') = \phi_d(x)^{\mathrm{T}} \phi_d(x')$. According to Moore-Aronszajn theorem (Aronszajn, 1950), $k_d$ is a reproducing kernel that uniquely corresponds to a finite-dimensional RKHS $\mathcal{H}_d$. This dimensional reduction enables direct computation of (2) as $\Psi \Phi^{\mathrm{T}} \left( \Phi \Phi^{\mathrm{T}} + n\lambda I \right)^{-1}$ while significantly reducing the computational complexity from $O(n^3)$ to $O(nd^2 + d^3)$.

This finite-dimensional replacement causes $\mathcal{H}_d$ to lose the favorable theoretical properties. Specifically: (1) the universal approximation capability (limiting its ability to model complex dependencies between input and output), and (2) the characteristic property (leading to entangled conditional distributions given distinct inputs).

## 3.2. A Data-Adaptive Neural Feature Map

To alleviate the shortcomings mentioned above, we design a data-adaptive neural feature map $\phi_\theta : \mathcal{X} \to \mathbb{R}^d$ for the input space, which is inspired by Xu et al. (2021), originally developed for instrumental variable regression.

Specifically, Grünewälder et al. (2012) have proven that estimating the CME is equivalent to solving the following

function-valued ridge regression problem:

$$\underset{U:\mathcal{H}\to\mathcal{G}}{\arg\min}\ \frac{1}{n}\sum_{i=1}^{n}\|\psi(y_i)-U\phi(x_i)\|_{\mathcal{G}}^2+\lambda\|U\|_{\mathrm{HS}}^2. \quad (3)$$

Substituting $\phi$ with $\phi_\theta$ in Equation (3) yields a joint optimization over $\theta$ and $U$. With $U$ admitting a closed-form solution, we can optimize $\theta$ directly via minimizing the following loss function (see Appendix B for derivation):

$$\mathcal{L}(\theta)=-\mathrm{tr}\left(K_Y\Phi_\theta^{\mathrm{T}}\left(\Phi_\theta\Phi_\theta^{\mathrm{T}}+n\lambda I\right)^{-1}\Phi_\theta\right), \quad (4)$$

where $K_Y=\Psi^{\mathrm{T}}\Psi\in\mathbb{R}^{n\times n}$ is the output Gram matrix. A well-trained $\phi_\theta$ allows us to estimate the empirical CME as:

$$\begin{aligned}\widehat{\mathcal{U}}&=\Psi\Phi_\theta^{\mathrm{T}}\left(\Phi_\theta\Phi_\theta^{\mathrm{T}}+n\lambda I\right)^{-1}\\&=\Psi Q=\sum_{i=1}^{n}\psi(y_i)q_i,\end{aligned} \quad (5)$$

with $Q=\Phi_\theta^{\mathrm{T}}\left(\Phi_\theta\Phi_\theta^{\mathrm{T}}+n\lambda I\right)^{-1}\in\mathbb{R}^{n\times d}$, and $q_i\in\mathbb{R}^{1\times d}$ is the $i$-th row of $Q$. The vector $q_i:\mathbb{R}^d\to\mathbb{R}$ and $\psi(y_i)\in\mathcal{G}$ combine the operator $\psi(y_i)q_i:\mathcal{H}_\theta\to\mathcal{G}$, since $\mathcal{H}_\theta\subseteq\mathbb{R}^d$.

### 3.3. The Effectiveness of the Neural Feature Map

Through training, $\phi_\theta$ maps inputs into a data-adaptive finite-dimensional RKHS $\mathcal{H}_\theta$ that is maximally correlated with the output RKHS $\mathcal{G}$. As a result, complex dependencies between inputs and outputs can be effectively captured, mitigating the lack of the universal approximation property. Simultaneously, $\phi_\theta$ mines discriminative input-output mapping patterns from data, allowing conditional distributions given different inputs to be distinguished, thereby alleviating the absence of the characteristic property.

Additionally, when estimating $\widehat{\mathcal{U}}$ on finite samples, compared to the original $\mathcal{H}$, on the one hand $\mathcal{H}_\theta$ enjoys a lower risk of overfitting due to its bounded representational capacity, while $\mathcal{H}$ tends to fit noise; on the other hand, the parameterized nature of $\phi_\theta$ enables flexible feature learning that adapts to data-specific patterns, reducing underfitting potential. This dual regularization mechanism gives $\mathcal{H}_\theta$ better generalization performance than $\mathcal{H}$.

Moreover, such a data-adaptive feature map $\phi_\theta$ eliminates the need for choosing specific kernel functions, after all, a certain kernel is not always applicable to all tasks, even when they possess universal approximation capabilities. Infinite-dimensional kernels show remarkable sensitivity to the regularization parameter $\lambda$, and different tasks can exhibit varying sensitivities, making hyperparameter tuning extremely challenging, while the neural kernels maintain stable performance across parameter variations.

### 3.4. Finding the Privacy-Preserving Reduced Set

To preserve data privacy, we generate a reduced set approximating the original $\widehat{\mathcal{U}}$. This requires first establishing an assumption and deriving a corollary.

**Assumption 3.1.** Assume that feature maps $\phi_\theta$ and $\psi$ are bounded, i.e., there exist constants $B_X, B_Y > 0$, such that:

$$\sup_{x\in\mathcal{X}}\|\phi_\theta(x)\|\leqslant B_X,\quad\sup_{y\in\mathcal{Y}}\|\psi(y)\|_{\mathcal{G}}\leqslant B_Y.$$

**Corollary 3.2.** *Under Assumption 3.1, for all $i\in[n]$, weight vector $q_i$ in Equation (5) satisfies $\|q_i\|\leqslant\frac{B_X}{n\lambda}$.*

Corollary 3.2 indicates that we require a reduced set $V=\{v_j\}_{j=1}^m\subseteq\mathcal{Y}$ with bounded-weight matrix $R\in\mathbb{R}^{m\times d}$, where $m\ll n$, to approximate $\widehat{\mathcal{U}}=\Psi Q$. This leads to the following constrained vector-weighted low-rank kernel approximation problem:

$$\begin{aligned}(V,R)=\underset{V\subseteq\mathcal{Y},R\in\mathbb{R}^{m\times d}}{\arg\min}\|\Psi Q-\Psi_V R\|_{\mathrm{HS}}^2,\\\text{subject to }\|r_j\|\leqslant\frac{B_X}{m\lambda},\forall j\in[m].\end{aligned} \quad (6)$$

To solve this problem, we propose an iterative algorithm based on the Frank-Wolfe algorithm (Wolfe, 1976). We first empty $V_0$ and $R_0$, then at the $t$-th iteration:

$$v_{t+1}=\underset{v\in\mathcal{Y}}{\arg\max}\|h_t(v)\|^2, \quad (7)$$

$$r_{t+1}=\frac{B_X}{\lambda}\cdot\frac{h_t(v_{t+1})}{\|h_t(v_{t+1})\|}, \quad (8)$$

$$V_{t+1}=V_t\cup\{v_{t+1}\}, R_{t+1}=\begin{pmatrix}R_t\\r_{t+1}\end{pmatrix}, \quad (9)$$

where function $h_t:\mathcal{Y}\to\mathbb{R}^{1\times d}$ is defined as:

$$h_t(v)=\sum_{i=1}^{n}q_ik_{\mathcal{Y}}(y_i,v)-\frac{1}{t}\sum_{j=1}^{t}r_jk_{\mathcal{Y}}(v_j,v), \quad (10)$$

with the convention that the second term vanishes when $t=0$. To solve Equation (7), we can either perform exhaustive search over $\mathcal{Y}$ if it is discrete, or apply gradient ascent if $\mathcal{Y}$ is continuous and $k_{\mathcal{Y}}$ is smooth. After $m$ iterations, we set $V=V_m$ and $R=R_m/m$, thus defining the RNCME specification as $\widetilde{\mathcal{U}}=\Psi_V R$.

**Proposition 3.3.** *The set $V$ and matrix $R$ generated by the above iterative algorithm approximate the minimizer of the optimization problem (6).*

Proposition 3.3 indicates that, after $m$ iterations, we obtain the RNCME $\widetilde{\mathcal{U}}$ approximates the empirical CME $\widehat{\mathcal{U}}$. To assess the overall fidelity of $\widetilde{\mathcal{U}}$ as an estimator of the true CME $\mathcal{U}$, we decompose the error as follows:

$$\left\|\mathcal{U}-\widetilde{\mathcal{U}}\right\|_{\mathrm{HS}}\leqslant\underbrace{\left\|\mathcal{U}-\widehat{\mathcal{U}}\right\|_{\mathrm{HS}}}_{\text{Statistical error}}+\underbrace{\left\|\widehat{\mathcal{U}}-\widetilde{\mathcal{U}}\right\|_{\mathrm{HS}}}_{\text{Reduction error}}. \quad (11)$$

The first term measures how well $\widehat{\mathcal{U}}$ approximates $\mathcal{U}$, and the second term measures how well $\widetilde{\mathcal{U}}$ approximates the $\widehat{\mathcal{U}}$. The following theorem combines these two sources of error and provides a unified upper bound.

**Theorem 3.4.** *Let $\widetilde{\mathcal{U}}$ be the RNCME obtained after $m$ iterations of the reduction algorithm, then in probability,*

$$\left\| \mathcal{U} - \widetilde{\mathcal{U}} \right\|_{\mathrm{HS}} = O_p \left( \frac{1}{\sqrt{n\lambda}} + \sqrt{\lambda} + \frac{1}{\lambda} \sqrt{\frac{\ln m}{m}} \right).$$

*Proof Sketch.* We decompose the total error into statistical error and reduction error, as shown in Equation (11). The statistical error bound follows from standard CME convergence results (Song et al., 2009), yielding the probabilistic bound $O_p(1/\sqrt{n\lambda} + \sqrt{\lambda})$. The reduction error originates from applying Frank-Wolfe algorithm to optimize a convex function with curvature $O(1/\lambda^2)$ over a convex domain of radius $O(1/\lambda)$. By setting the Frank-Wolfe step size to $1/(t+1)$, we obtain a recurrence relation (Jaggi, 2013) leading to the reduction error bound $O(\sqrt{\ln m/m}/\lambda)$. Combining these two independent bounds via the triangle inequality completes the proof.

The details, including the specifics about the Frank-Wolfe algorithm, and all the proofs can be found in Appendix C.

### 3.5. Learnware with RNCME

The RNCME $\widetilde{\mathcal{U}}$ effectively captures input-output mapping patterns while preserving data privacy, making it a promising specification. We demonstrate how RNCME enhances the learnware framework introduced in Section 2.2.

**The Submitting Stage** Consider a developer $c$ who trains a well-performed model $f_c : \mathcal{X}_c \rightarrow \mathcal{Y}_c$ on local data $D_c$. Using $D_c$, the developer locally trains the neural feature map $\phi_{\theta_c} : \mathcal{X}_c \rightarrow \mathbb{R}^d$ and generates the RNCME specification $\widetilde{\mathcal{U}}_c$. Finally, $f_c$ and $\widetilde{\mathcal{U}}_c$ are submitted to the market.

**The Deploying Stage** A user with task $t$ holds small labeled data $D_l$ and large unlabeled data $D_u$. The user likewise trains the neural feature map $\phi_{\theta_t}$ on $D_l$ to generates the RNCME specification $\widetilde{\mathcal{U}}_t$. Upon receiving $\widetilde{\mathcal{U}}_t$, the market compares it with all learnware specifications $\widetilde{\mathcal{U}}_c$ and returns the most relevant learnware $f_{c^*}$ for processing $D_u$, where:

$$c^* = \underset{c \in [N]}{\arg\min} \left\| \widetilde{\mathcal{U}}_c - \widetilde{\mathcal{U}}_t \right\|_{\mathrm{HS}}^2. \tag{12}$$

We provide pseudocode for all stages of the RNCME-based learnware framework in Appendix D, including the training of neural feature maps, generation of the reduced set, the submission process, and the deployment process.

## 4. Experiments

We conduct experiments to evaluate the effectiveness and efficiency of our proposed RNCME method, which are organized into: real-world regression tasks, real-world classification tasks, and comprehensive analyses including time efficiency, reduced set generation, and neural feature map training. For brevity, we defer the detailed experimental setup (e.g., hyperparameters) to Appendix E.

### 4.1. Real-World Regression Experiments

We first conduct regression experiments on a small real-world California Housing Prices dataset (Nugent, 2017), which contains 20,640 samples with 10 features.

**Experimental Details** We consider three features as potential outputs and the remaining seven features as potential inputs. From these, we select five input features and one label feature to generate $\binom{7}{5} \cdot \binom{3}{1} = 63$ distinct tasks. These tasks are divided into 42 tasks for constructing the learnware market and 21 tasks for simulating user scenarios. For each learnware task, we create a learnware by training a Gradient Boosted Decision Tree (GBDT) (Friedman, 2001) and generating its corresponding specification. Then, each user dataset is split into two disjoint parts: the testing part, which contains 90% of the samples and is used to evaluate model performance; and the specification part, which contains the remaining 10% and is used to generate the specification. In RNCME, $\phi_\theta$ is implemented as a single layer neural network. We compare our RNCME method with the basic RKME method (Wu et al., 2023), since all labeled-RKME improvements are constrained to classification tasks.

**Evaluation Metric** We employ the Normalized Discounted Cumulative Gain (NDCG) metric to evaluate the recommendation quality of each method. For each user task, we recommend an ordered set of $k$ learnwares $\{f_{\mathcal{I}_1}, f_{\mathcal{I}_2}, \cdots, f_{\mathcal{I}_k}\}$ based on specifications, where $f_{\mathcal{I}_i}$ is the $i$-th nearest learnware to the user task. We define the relevance score $rel_i$ of $f_{\mathcal{I}_i}$ as the proportion of candidate learnwares it outperforms on the user task. Formally, if $\mathcal{J}_i$ denotes the performance rank of $f_{\mathcal{I}_i}$ on the user task, then $rel_i = \frac{N - \mathcal{J}_i}{N-1}$. In the ideal ranking, the $i$-th best-performing learnware receives an ideal relevance score of $rel_i^* = \frac{N-i}{N-1}$. The NDCG@$k$ is then computed as:

$$\mathrm{NDCG}@k = \frac{\sum_{i=1}^k \frac{rel_i}{\log_2(i+1)}}{\sum_{i=1}^k \frac{rel_i^*}{\log_2(i+1)}}. \tag{13}$$

NDCG@$k$ is in range $[0, 1]$, and a larger NDCG@$k$ indicates better agreement between the specification-based recommendation and the actual performance. Finally, we compute the average NDCG@$k$ over all user tasks to evaluate the overall performance of each method.

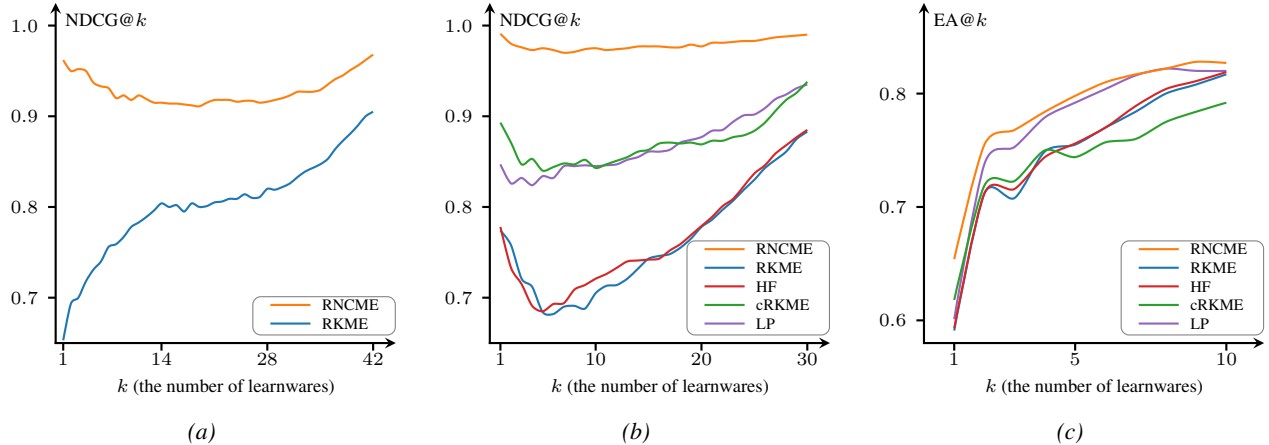

*Figure 1.* Experimental results on real-world regression and classification datasets: (a) NDCG@$k$ performance on California Housing Prices dataset across all $k \leqslant n$; (b) NDCG@$k$ performance on NICO$^{++}$ dataset across all $k \leqslant n$; (c) the average ensemble accuracy for top-$k$ recommended learnwares on NICO$^{++}$ dataset across $k \leqslant 10$.

**Results** We present the experimental regression results in Figure 1a, showing that RNCME achieves superior performance compared to RKME for every $k$.

### 4.2. Real-World Classification Experiments

To further validate the effectiveness of our RNCME method, we conduct experiments on a larger and more complex real-world NICO$^{++}$ dataset (Zhang et al., 2023), which is designed for OOD image classification. The dataset contains over 0.2M images across 80 categories, with each image annotated by category label (e.g., cat or dog) and domain label (e.g., lying or running). The dataset provides 10 aligned common domains shared across all categories and 10 unique domains specific to each category, effectively simulating real-world scenarios with potential distribution shifts between training and testing data. This characteristic makes it suitable for evaluating the learnware paradigm.

**Experimental Details** We construct the learnware market using the common-domain portion of the NICO$^{++}$ dataset. First, we create 30 tasks by first randomly selecting 30 to 40 categories per task, and then sampling 90% of images from each selected category. For each task, we train a neural network that uses a frozen DenseNet201 (Huang et al., 2017) backbone, followed by a two-layer MLP as the classifier. Then we generate the specification of the trained model to form a learnware. To simulate user scenarios with inherent domain shifts, we create 20 user tasks using the unique-domain portion of the NICO$^{++}$ dataset. Each task is divided into a testing subset to evaluate learnwares and a specification subset to generate the specification. The neural feature map $\phi_\theta$ is implemented as a single-layer neural network that takes DenseNet201-extracted features as input. Except for the basic RKME method, we also compare our

RNCME method with Heterogeneous Features (HF) (Tan et al., 2024a), class-wise RKME (cRKME) (Liu et al., 2025) and Linear Proxy (LP) (Guo et al., 2023).

**Evaluation Metrics** Except for the NDCG@$k$ metric defined in Equation (13), we also report the Ensemble Accuracy (EA) of the top-$k$ recommended learnwares, where the ensemble is performed via soft voting. And similarly we compute the average EA@$k$ over all user tasks to evaluate the overall performance of each method. Different from NDCG@$k$, we only report the average ensemble accuracy for $k \leqslant 10$, since ensembling all learnwares is less meaningful in practice.

**Results** The NDCG@$k$ and EA@$k$ results are shown in Figures 1b and 1c, respectively. These results show that RNCME achieves superior performance compared to any other method for every $k$ value on both NDCG@$k$ and EA@$k$ metrics. Particularly, for $k = 1$, there are $16/20$ tasks received the best-performing learnware, and all received top-3 learnwares.

### 4.3. Computational Efficiency Evaluation

To demonstrate the efficiency of RNCME, especially compared to CME with infinity-dimensional kernel, we conduct experiments on a dataset of $n = 40,000$ samples from NICO$^{++}$. In this part, CME uses RBF kernels, has $O(n^3)$ estimation computational complexity. RNCME is further split into classification and regression, since the solutions to Equation (7) differ. For classification, $v_{t+1}$ is obtained by searching a fixed label space $\mathcal{Y}$, allowing for extensive pre-computation. However, regression requires gradient ascent with backpropagation to estimate $v_{t+1}$, making it significantly more computationally intensive.

*Table 1.* Time consumption (in seconds, mean $\pm$ std) of each method at different stages, where "Cls." and "Reg." denote classification and regression, respectively.

| Method | Generation | Compression | Comparison |
|--------|-----------|-------------|------------|
| RKME | 0.169 $\pm 0.008$ | 14.510 $\pm 0.233$ | $< 0.001$ |
| RNCME$_{Cls.}$ | 0.693 $\pm 0.009$ | 0.516 $\pm 0.021$ | $< 0.001$ |
| RNCME$_{Reg.}$ | 0.583 $\pm 0.008$ | 26.648 $\pm 0.549$ | $< 0.001$ |
| CME | 13.232 $\pm 0.183$ | $> 10,000$ | 0.022 $\pm 0.001$ |
| HF | 0.212 $\pm 0.020$ | 13.213 $\pm 0.323$ | 0.042 $\pm 0.019$ |
| cRKME | 0.223 $\pm 0.019$ | 15.630 $\pm 0.234$ | 0.479 $\pm 0.033$ |
| LP | 0.192 $\pm 0.011$ | 23.402 $\pm 0.932$ | 0.089 $\pm 0.020$ |

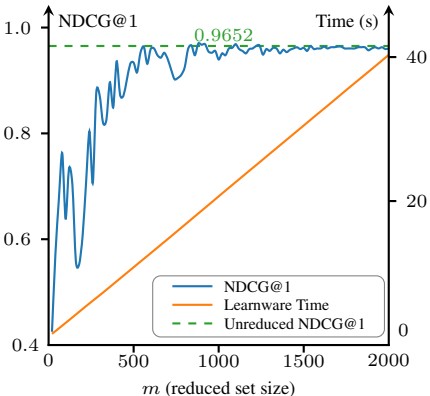

*Figure 2.* The time cost and recommendation performance (NDCG@1) of RNCME with varying reduced set size $m$ on the California Housing Prices Dataset.

We report in Table 1 the time required by each method for the following three stages: (1) **Generation** of the original specifications; (2) **Compression** to obtain privacy-preserving specifications (e.g., reduced sets or linear proxies); and (3) **Comparison** between a user task and a single learnware. Note that for RNCME, the generation stage includes an extra training stage of the neural feature map $\phi_\theta$. In the compression stage, the reduced set size is set to $m = 1,000$. All results are reported as mean $\pm$ standard deviation over repeated experiments.

From Table 1, we observe that RNCME significantly reduces the computational time compared to CME, demonstrating a complexity reduction from $O(n^3)$ to $O(nd^2 + d^3)$. Specifically, CME did not complete the compression stage within 10,000 seconds, therefore the comparison time of CME is measured on a subset of size $m$ directly. In contrast, RNCME finishes the compression stage in only 0.51 seconds for classification tasks and 26.64 seconds for regression tasks. Moreover, both RKME and RNCME complete a single comparison within 0.001 seconds, while cRKME takes 0.47 seconds. This efficiency gap becomes even more pronounced when the number of comparisons scales up in practical applications.

### 4.4. Impact of Reduced Set Size

To investigate the trade-off between efficiency and effectiveness in our reducing algorithm proposed in Section 3.4, we vary the reduced set size $m$ and measure its impact on runtime and NDCG@1. This allows us to examine how compression influences computational cost and ranking quality. We conduct this analysis on the California Housing Prices dataset, focusing on regression tasks due to their typically higher computational demands than classification tasks.

The experimental results are shown in Figure 2. From the figure, we can observe that as the reduced set size $m$ increases, the time required to generate the reduced set grows almost linearly, which is consistent with the iterative nature of our algorithm. Meanwhile, the NDCG@1 metric of RNCME exhibits a general trend of improvement and eventual convergence as $m$ increases. When $m$ exceeds approximately 800, the NDCG value converges to and even surpasses the performance achieved on the original full dataset, while the total reduction time remains within an acceptable range (on the order of tens of seconds). These results indicate the effectiveness of our proposed reduction algorithm.

### 4.5. Ablation Study on Neural Feature Map

To address the degradation of favorable properties resulting from finite-dimensional kernel replacement (discussed in Section 3.1), we introduce a trainable neural feature map $\phi_\theta$ designed to better capture input-output mapping patterns. This approach is elaborated in Section 3.2. We validate its effectiveness through an ablation study on a synthetic complex dataset, since existing real-world datasets lack the complexity needed to properly stress-test our method.

Specifically, we construct a hand-crafted, highly complex conditional distribution $P_{Y|X}$ (see Section E.6) and sample a dataset of 12,000 instances from it. Of these, $n_1 = 10,000$ samples are used to train the neural feature map $\phi_\theta$ and estimate CME $\widehat{\mathcal{U}}$. The remaining $n_2 = 2,000$ samples $\{(x_j, y_j)\}_{j=1}^{n_2}$ serve as the test set for evaluating the quality of the estimated CME. For test inputs $\{x_j\}_{j=1}^{n_2}$ we use $\widehat{\mathcal{U}}$ to predict the KME of $P_Y$, and record the mean MMD between the predicted and true KMEs:

$$\frac{1}{n_2^2} \left\| \sum_{j=1}^{n_2} \widehat{\mathcal{U}}_{Y|X=x_j} - \sum_{j=1}^{n_2} \psi(y_j) \right\|_{\mathcal{G}}^2.$$

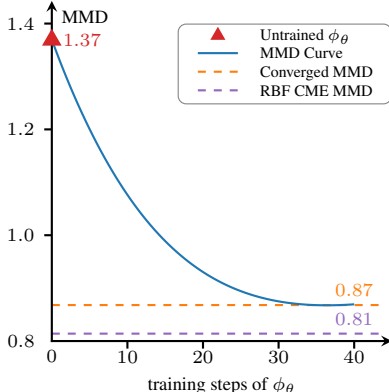

*Figure 3.* MMD from estimated to true CMEs over training steps of neural feature map $\phi_\theta$ on a complex synthetic dataset.

We employ a neural feature map $\phi_\theta$ consisting of a linear layer followed by a sigmoid activation, trained with a learning rate of $5 \times 10^{-4}$. To evaluate the impact of training, we examine the relationship between the number of training steps (where step 0 corresponds to the untrained $\phi_\theta$) and the resulting MMD. As shown in Figure 3, the trained $\phi_\theta$ reduces the MMD from 1.37 (untrained baseline) to 0.87, while the RBF-kernel CME achieves 0.81. This corresponds to an improvement of $\frac{1.37-0.87}{1.37-0.81} \approx 89.3\%$ relative to the gap between the untrained model and the RBF-kernel baseline. These results demonstrate that training $\phi_\theta$ effectively captures complex conditional dependencies.

## 5. Related Work

**Learnware**  The learnware paradigm (Zhou, 2016; Zhou & Tan, 2024) aims to establish a market of learnwares that help users identify and reuse helpful machine learning model(s) instead of training from scratch. The specification plays a crucial role in the learnware paradigm, and there have been some efforts in this direction. Wu et al. (2023) proposed the RKME specification, which uses a reduced set to approximate the empirical KME of the training data, while Lei et al. (2024) demonstrated its capability to preserve data privacy. Liu et al. (2024) proposed an evolvable learnware specification to address the challenge of evaluating a model's capacity to exceed its original training task. Tan et al. (2024a) adapted RKME for heterogeneous feature spaces. Guo et al. (2023); Tan et al. (2024a); Liu et al. (2025) leveraged label information to model the input-output conditional distributions. Shen & Li (2025) proposed a learnable specification, providing more flexible representations of task distributions. Recently, PAVE specifications (Tan et al., 2025; Shi et al., 2026; Liu et al., 2026) represent model functionalities in a shared parameter space by leveraging task vectors (Ilharco et al., 2023). Based on these works, Beimingwu (Tan et al., 2024b) has been released as the first learnware dock system.

**Kernel Mean Embedding**  Kernel Mean Embedding (KME) (Smola et al., 2007) offers a principled nonparametric framework for embedding probability distributions into RKHS. This approach was further advanced by Gretton et al. (2012), who established its theoretical foundations and demonstrated its intimate connection to Maximum Mean Discrepancy (MMD), providing a powerful tool for two-sample testing and distribution comparison. A key strength of KME is its ability to represent entire distributions as single points in an RKHS, enabling efficient computation of distances, expectations, and similarities between distributions without requiring explicit density estimation. This line of work was subsequently extended to conditional distributions by Song et al. (2009), leading to the development of Conditional Mean Embedding (CME). Notably, Grünewälder et al. (2012) demonstrated that empirical CME estimation can be equivalently formulated as a vector-valued ridge regression problem, bridging kernel methods with classical regression techniques. More recently, Shimizu et al. (2024) enhanced CME by integrating deep neural networks, combining the representation power of deep learning with the theoretical soundness of kernel embeddings. For a comprehensive treatment of these developments and their wide-ranging applications, Muandet et al. (2017) offers a detailed survey.

## 6. Conclusion

The learnware paradigm aims to establish a market of learnwares, enabling users to effectively identify and reuse relevant models via specifications instead of training from scratch. To address the limitations of existing labeled-RKME improvements, i.e., the lack clear theoretical explanations and restriction to classification tasks, we propose a novel specification called RNCME. Our method directly maps input-output conditional distributions into RKHS through CME, thereby inheriting CME's theoretical foundations while supporting both regression and classification tasks. To overcome the high computational bottleneck of standard CME, we propose a data-adaptive neural feature map for the input space, which significantly reduces the estimation complexity and avoids the need for manual kernel selection. Through a dedicated training phase, the neural feature map captures essential input-output mapping patterns, mitigating the limitations of finite-dimensional RKHSs in terms of universal approximation capability and characteristic properties. Furthermore, we develop a privacy-preserving iterative algorithm for constrained vector-weighted low-rank kernel approximation, which generates a compact reduced set that approximates the empirical CME. Experimental results show that RNCME effectively generates specifications that accurately capture input-output mapping patterns, leading to efficient and accurate learnware recommendation for both regression and classification tasks.

## Acknowledgements

We thank Zhi-Hao Tan for his helpful discussions and suggestions. This work was supported by the Major Program (JD) of Hubei Province (2023BAA024), the Fundamental and Interdisciplinary Disciplines Breakthrough Plan of the Ministry of Education of China (JYB2025XDXM118) and the "111 Center" (B26023).

## Impact Statement

This paper presents work whose goal is to advance the field of machine learning. There are many potential societal consequences of our work, none of which we feel must be specifically highlighted here.

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

# A. Proof of Equivalence Between the Empirical and Original CME

In order to prove the equivalence between Equations (1) and (2), we first introduce two lemmas.

**Lemma A.1.** *For any operator $C : \mathcal{H} \to \mathcal{H}$, we have $\left(\Psi \otimes \Phi^{\mathrm{T}}\right) C = \Psi \Phi^{\mathrm{T}} C$.*

*Proof.* Here, $\otimes$ is the tensor product operator. The tensor product space $\mathcal{G} \otimes \mathcal{H}$ is isomorphic to the Hilbert-Schmidt space $\mathrm{HS}(\mathcal{H}, \mathcal{G})$, and the isomorphic mapping $\iota : \mathcal{G} \otimes \mathcal{H} \to \mathrm{HS}(\mathcal{H}, \mathcal{G})$ is defined as:

$$\forall g \in \mathcal{G}, h \in \mathcal{H}, \iota(g \otimes h) = \langle h, \cdot \rangle_{\mathcal{H}} \cdot g.$$

Therefore, $\forall h \in \mathcal{H}, C(h) \in \mathcal{H}$, we have:

$$
\begin{aligned}
\left(\Psi \otimes \Phi^{\mathrm{T}}\right) C(h) &= \sum_{i=1}^{n} (\psi(y_i) \otimes \phi(x_i)) C(h) \\
&= \sum_{i=1}^{n} \psi(y_i) \cdot \langle \phi(x_i), C(h) \rangle_{\mathcal{H}} \\
&= \sum_{i=1}^{n} \psi(y_i) \cdot \left(\Phi^{\mathrm{T}} C(h)\right)_i \\
&= \Psi \left(\Phi^{\mathrm{T}} C(h)\right) = \Psi \Phi^{\mathrm{T}} C(h).
\end{aligned}
$$

$\square$

**Lemma A.2.** $\Phi^{\mathrm{T}} \left(\Phi \otimes \Phi^{\mathrm{T}}\right) = \Phi^{\mathrm{T}} \Phi \Phi^{\mathrm{T}}$.

*Proof.* Let $\mathcal{I}$ denote the identity operator on $\mathcal{H}$. Operator composition is right-associative, then according to Lemma A.1:

$$\Phi^{\mathrm{T}} \left(\Phi \otimes \Phi^{\mathrm{T}}\right) = \Phi^{\mathrm{T}} \left(\Phi \otimes \Phi^{\mathrm{T}}\right) \mathcal{I} = \Phi^{\mathrm{T}} \Phi \Phi^{\mathrm{T}} \mathcal{I} = \Phi^{\mathrm{T}} \Phi \Phi^{\mathrm{T}}.$$

$\square$

**The Proof of Equivalence**

*Proof.* Let $\Psi \otimes \Phi^{\mathrm{T}}$ denote $\sum_{i=1}^{n} \psi(y_i) \otimes \phi(x_i)$ and $\Phi \otimes \Phi^{\mathrm{T}}$ denote $\sum_{i=1}^{n} \phi(x_i) \otimes \phi(x_i)$. According to Lemma A.1, we can rewrite Equation (2) as:

$$\widehat{\mathcal{U}} = \left(\frac{1}{n} \Psi \otimes \Phi^{\mathrm{T}}\right) \left(\frac{1}{n} \Phi \otimes \Phi^{\mathrm{T}} + \lambda \mathcal{I}\right)^{-1} = \Psi \Phi^{\mathrm{T}} \left(\Phi \otimes \Phi^{\mathrm{T}} + n\lambda \mathcal{I}\right)^{-1},$$

where $\mathcal{I}$ is the identity operator on $\mathcal{H}$. Therefore we need to prove:

$$\Phi^{\mathrm{T}} \left(\Phi \otimes \Phi^{\mathrm{T}} + n\lambda \mathcal{I}\right)^{-1} = \left(\Phi^{\mathrm{T}} \Phi + n\lambda I\right)^{-1} \Phi^{\mathrm{T}}, \tag{14}$$

where $\Phi^{\mathrm{T}} \Phi = K_X$ is the Gram matrix, and $I$ is the identity matrix. According to the linearity, the distributive property and Lemma A.2, we have:

$$
\begin{aligned}
\Phi^{\mathrm{T}} \left(\Phi \otimes \Phi^{\mathrm{T}} + n\lambda \mathcal{I}\right) &= \Phi^{\mathrm{T}} \left(\Phi \otimes \Phi^{\mathrm{T}}\right) + n\lambda \Phi^{\mathrm{T}} \\
&= \Phi^{\mathrm{T}} \Phi \Phi^{\mathrm{T}} + n\lambda \Phi^{\mathrm{T}} \\
&= \left(\Phi^{\mathrm{T}} \Phi + n\lambda I\right) \Phi^{\mathrm{T}} \\
\implies \Phi^{\mathrm{T}} &= \left(\Phi^{\mathrm{T}} \Phi + n\lambda I\right) \Phi^{\mathrm{T}} \left(\Phi \otimes \Phi^{\mathrm{T}} + n\lambda \mathcal{I}\right)^{-1} \\
\implies \left(\Phi^{\mathrm{T}} \Phi + n\lambda I\right)^{-1} \Phi^{\mathrm{T}} &= \Phi^{\mathrm{T}} \left(\Phi \otimes \Phi^{\mathrm{T}} + n\lambda \mathcal{I}\right)^{-1}
\end{aligned}
$$

which is equivalent to Equation (14). Therefore, Equations (1) and (2) are equivalent. $\square$

# B. Derivation of the Loss Function

Similarly, in order to derive the loss function, we first introduce a lemma.

**Lemma B.1.** *The closed-form solution of Equation (3) is:*

$$U^* = \Psi \otimes \Phi^{\mathrm{T}} \left( \Phi \otimes \Phi^{\mathrm{T}} + n\lambda\mathcal{I} \right)^{-1}$$

*Proof.* Let $\delta U$ be a small perturbation of $U$, then the Fréchet derivative of $\mathcal{L}_{\mathrm{ridge}}(U)$ at $U$ is given by:

$$\delta\mathcal{L}_{\mathrm{ridge}}(U) = -\frac{2}{n}\sum_{i=1}^{n}\langle\psi(y_i) - U\phi(x_i), \delta U\phi(x_i)\rangle_{\mathcal{G}} + 2\lambda\langle U, \delta U\rangle_{\mathrm{HS}}$$

$$= \left\langle -\frac{2}{n}\sum_{i=1}^{n}(\psi(y_i) - U\phi(x_i)) \otimes \phi(x_i) + 2\lambda U, \delta U \right\rangle_{\mathrm{HS}}.$$

By definition of the Fréchet derivative, we identify the gradient:

$$\nabla\mathcal{L}_{\mathrm{ridge}}(U) = -\frac{2}{n}\sum_{i=1}^{n}(\psi(y_i) - U\phi(x_i)) \otimes \phi(x_i) + 2\lambda U.$$

Setting the gradient to zero yields the optimality condition:

$$n\lambda U^* = \sum_{i=1}^{n}(\psi(y_i) - U^*\phi(x_i)) \otimes \phi(x_i) = \Psi \otimes \Phi^{\mathrm{T}} - U^*(\Phi \otimes \Phi^{\mathrm{T}}),$$

which means $U^* = \Psi \otimes \Phi^{\mathrm{T}} \left( \Phi \otimes \Phi^{\mathrm{T}} + n\lambda\mathcal{I} \right)^{-1}$. $\qquad\square$

**The Derivation**

We can rewrite Equation (3) as:

$$\mathcal{L}_{\mathrm{ridge}}(U) = \frac{1}{n}\|\Psi - U\Phi\|_{\mathcal{G}}^2 + \lambda\|U\|_{\mathrm{HS}}^2.$$

When we use the neural feature map $\phi_\theta$, we have the explicit feature matrix $\Phi_\theta \in \mathbb{R}^{d\times n}$ (simply denoted as $\Phi$ here). When $\Phi$ is finite-dimensional, the tensor product is equivalent to operator composition, i.e., we have $U^* = \Psi\Phi^{\mathrm{T}}\left(\Phi\Phi^{\mathrm{T}} + n\lambda I\right)^{-1}$.

Now we denote $A = \left(\Phi\Phi^{\mathrm{T}} + n\lambda I\right)^{-1} \in \mathbb{R}^{d\times d}$ and $B = \Phi^{\mathrm{T}}A\Phi \in \mathbb{R}^{n\times n}$, then the closed-form solution $U^*$ in Lemma B.1 can be rewritten as $U^* = \Psi\Phi^{\mathrm{T}}A$. Substituting $U^*$ into $\mathcal{L}_{\mathrm{ridge}}(U)$, we have:

$$\begin{aligned}
n\cdot\mathcal{L}_{\mathrm{ridge}}(\theta) &= \|\Psi(I - B)\|_{\mathcal{G}}^2 + n\lambda\left\|\Psi\Phi^{\mathrm{T}}A\right\|_{\mathcal{G}}^2 \\
&= \mathrm{tr}\left((I - B)^{\mathrm{T}}K_Y(I - B)\right) + n\lambda\cdot\mathrm{tr}\left(A^{\mathrm{T}}\Phi K_Y\Phi^{\mathrm{T}}A\right) \\
&= \mathrm{tr}\left(K_Y(I - B)(I - B)^{\mathrm{T}}\right) + n\lambda\cdot\mathrm{tr}\left(K_Y\Phi^{\mathrm{T}}AA^{\mathrm{T}}\Phi\right) \\
&= \mathrm{tr}(K_Y(I - B)) - \mathrm{tr}\left(K_Y\left((I - B)B^{\mathrm{T}} - n\lambda\Phi^{\mathrm{T}}AA^{\mathrm{T}}\Phi\right)\right).
\end{aligned}$$

Matrices $A$ and $B$ are both symmetric, so we consider the last term:

$$\begin{aligned}
(I - B)B^{\mathrm{T}} - n\lambda\Phi^{\mathrm{T}}AA^{\mathrm{T}}\Phi &= \Phi^{\mathrm{T}}A\Phi - \Phi^{\mathrm{T}}A\Phi\Phi^{\mathrm{T}}A\Phi - n\lambda\Phi^{\mathrm{T}}AA^{\mathrm{T}}\Phi \\
&= \Phi^{\mathrm{T}}\left(I - A\Phi\Phi^{\mathrm{T}} - n\lambda A\right)A\Phi \\
&= \Phi^{\mathrm{T}}\left(I - A\left(\Phi\Phi^{\mathrm{T}} + n\lambda I\right)\right)A\Phi \\
&= \Phi^{\mathrm{T}}\left(I - AA^{-1}\right)A\Phi = 0.
\end{aligned}$$

Thus the last term is zero, and we have:

$$\mathcal{L}_{\mathrm{ridge}}(\theta) = \frac{1}{n}\left(\mathrm{tr}(K_Y) - \mathrm{tr}(K_Y B)\right) \propto -\mathrm{tr}(K_Y B).$$

Therefore, the optimization of $\theta$ in Equation (3) is equivalent to minimizing the loss function:

$$\mathcal{L}(\theta) = -\mathrm{tr}\left(K_Y\Phi_\theta^{\mathrm{T}}\left(\Phi_\theta\Phi_\theta^{\mathrm{T}} + n\lambda I\right)^{-1}\Phi_\theta\right).$$

# C. The Details of the Reduced Set Generation

Given the estimation of empirical CME $\widehat{\mathcal{U}} = \Psi Q$ in Equation (5), we seek a reduced set $V = \{v_j\}_{j=1}^m \subseteq \mathcal{Y}$ with bounded-weight matrix $R \in \mathbb{R}^{m \times d}$ to approximate $\widehat{\mathcal{U}}$. We first prove Corollary 3.2, then introduce the Frank-Wolfe algorithm, and finally prove all theorems.

## C.1. The Proof of the Bounded-Weight Corollary

*Proof.* First, for a symmetric and positive definite matrix $A$, we have $\|A^{-1}\| = 1/\lambda_{\min}(A)$, where $\lambda_{\min}(A)$ denotes the minimal eigenvalue of $A$. Therefore, we need to compute the minimal eigenvalue of the symmetric and positive definite matrix $\Phi_\theta \Phi_\theta^T + n\lambda I$. Suppose the minimal eigenvalue of $\Phi_\theta \Phi_\theta^T$ is $\lambda_d$, and since $\Phi_\theta \Phi_\theta^T$ is postive semi-definite, we have:

$$\lambda_{\min}\left(\Phi_\theta \Phi_\theta^T + n\lambda I\right) = \lambda_d + n\lambda \geqslant n\lambda,$$

Then, under Assumption 3.1, by Cauchy-Schwarz inequality, we have:

$$\|q_i\| \leqslant \left\|\phi_\theta(x_i)^T\right\| \cdot \left\|\left(\Phi_\theta \Phi_\theta^T + n\lambda I\right)^{-1}\right\| \leqslant \frac{B_X}{n\lambda}.$$

$\square$

## C.2. Frank-Wolfe Algorithm

Frank-Wolfe algorithm (Wolfe, 1976) is an iterative first-order optimization algorithm for constrained convex optimization. Suppose $\mathcal{D}$ is a compact convex set in a vector space, and $f : \mathcal{D} \to \mathbb{R}$ is a convex, differentiable function. After initializing $x_0 \in \mathcal{D}$, Frank-Wolfe algorithm solves the optimization problem $\min_{x \in \mathcal{D}} f(x)$ by performing an iterative procedure, where in each iteration $t$, the following steps are taken:

$$s_{t+1} \in \arg\min_{s \in \mathcal{D}} \langle \nabla f(x_t), s \rangle, \tag{15}$$

$$x_{t+1} = (1 - \alpha_t)x_t + \alpha_t s_{t+1}, \tag{16}$$

where, $\alpha_t \in [0, 1]$ can either be simply set to $\alpha_t = 1/(t+1)$ or optimized via line search to find the point in the segment with optimal value. Equation (15) minimizes the linear approximation of the problem given by the first-order Taylor approximation of $f$ around $x_t$ constrained to stay within $\mathcal{D}$.

## C.3. The Proof of the Optimization Proposition

To prove Proposition 3.3, it suffices to demonstrate that the entire procedure follows the standard Frank-Wolfe framework.

*Proof.* Firstly, we construct a set $\mathcal{D}_0$ as:

$$\mathcal{D}_0 = \left\{ \psi(v)r : v \in \mathcal{Y}, r \in \mathbb{R}^{1 \times d}, \|r\| \leqslant \frac{B_X}{\lambda} \right\}. \tag{17}$$

Notably, $\|r\| \leqslant B_X/\lambda$ holds rather than $B_X/(m\lambda)$, as will be justified subsequently. Denoting $\mathcal{D}$ as the convex hull of $\mathcal{D}_0$, we define a convex differentiable function $f : \mathcal{D} \to \mathbb{R}$ as:

$$f(U) = \frac{1}{2} \left\|\widehat{\mathcal{U}} - U\right\|_{\mathrm{HS}}^2. \tag{18}$$

Our goal is $\min_{U \in \mathcal{D}} f(U)$. Following the Frank-Wolfe algorithm, we initialize $U_0$ as the zero vector in $\mathcal{D}$, and set $\alpha_t$ to $1/(t+1)$, which yields the following iterative procedure:

$$(v_{t+1}, r_{t+1}) \in \arg\min_{\psi(v)r \in \mathcal{D}} \langle \nabla f(U_t), \psi(v)r \rangle_{\mathrm{HS}}, \tag{19}$$

$$U_{t+1} = \frac{t}{t+1}U_t + \frac{\psi(v_{t+1})r_{t+1}}{t+1}. \tag{20}$$

Equation (20) gives $U_t = \frac{1}{t}\sum_{j=1}^{t}\psi(v_j)r_j \in \mathcal{D}$, which implies that after $m$ iterations, $\left\|\widehat{\mathcal{U}} - \Psi_{V_m}R_m/m\right\|_{\mathrm{HS}}^2$ is minimized. This explains why we set the final weight matrix as $R_m/m$ rather than $R_m$ itself, and why the constraint in the construction of $\mathcal{D}_0$ is set as $B_X/\lambda$ instead of $B_X/(m\lambda)$.

We then define $W_t = \widehat{\mathcal{U}} - U_t = -\nabla f(U_t)$, and its adjoint operator $W_t^* : \mathcal{G} \to \mathbb{R}^{1\times d}$ is defined by the relation $\langle W_t, \psi(v)r\rangle_{\mathrm{HS}} = \langle r, W_t^*\psi(v)\rangle$, which implies $\langle \nabla f(U_t), \psi(v)r\rangle_{\mathrm{HS}} = -\langle r, W_t^*\psi(v)\rangle$. The explicit form of $W_t^*\psi(v)$ can be derived from the definitions of $W_t$ and $U_t$:

$$W_t^*\psi(v) = \sum_{i=1}^{n} q_i k_{\mathcal{Y}}(y_i, v) - \frac{1}{t}\sum_{j=1}^{t} r_j k_{\mathcal{Y}}(v_j, v), \text{ for } t \geqslant 1.$$

For the base case $t = 0$ (where $U_0 = 0$), the second term is absent, making this expression consistent with the definition of $h_t(v)$ in Equation (10). Therefore, Equation (19) can be rewritten as:

$$(v_{t+1}, r_{t+1}) \in \underset{\psi(v)r\in\mathcal{D}}{\arg\max}\,\langle r, h_t(v)\rangle. \tag{21}$$

By the Cauchy-Schwarz inequality, we have $\langle r, h_t(v)\rangle \leqslant \|r\| \cdot \|h_t(v)\|$, with equality holding if and only if $r$ is collinear with $h_t(v)$. Under the constraint $\|r\| \leqslant B_X/\lambda$, the inner product is maximized by choosing $r$ to be aligned with $h_t(v)$ and of maximum permissible norm. Consequently, solving Equation (19) is equivalent to the following two-step procedure:

$$v_{t+1} = \underset{v\in\mathcal{Y}}{\arg\max}\,\|h_t(v)\|, \tag{22}$$

$$r_{t+1} = \frac{B_X}{\lambda} \cdot \frac{h_t(v_{t+1})}{\|h_t(v_{t+1})\|}. \tag{23}$$

They are equivalent to Equations (7) and (8). Therefore, our iterative algorithm conforms to the standard Frank-Wolfe framework and the generated $(V_m, R_m/m)$ approximately minimizes the optimization objective in Equation (6). $\qquad\square$

## C.4. Lemmas for Proving the Upper Bound Theorem

In order to prove Theorem 3.4, we first provide Lemmas C.1 to C.4.

**Lemma C.1.** *The optimization problem* $\min_{U\in\mathcal{D}} f(U)$ *has a trivial solution* $U^* = \widehat{\mathcal{U}}$, *where* $\mathcal{D}$ *is the convex hull of set* $\mathcal{D}_0$ *defined in Equation* (17)*, and* $f$ *is defined in Equation* (18)*.*

*Proof.* It is obvious that $\min_{U\in\mathcal{G}} f(U) = 0$ when $U = \widehat{\mathcal{U}}$, but it may not hold when the domain changes to $\mathcal{D}$, therefore we need to prove that $\widehat{\mathcal{U}} \in \mathcal{D}$. Since $\mathcal{D}$ is the convex hull of set $\mathcal{D}_0$ defined in Equation (17), we know that $\mathcal{D}$ contains all elements expressible as finite linear combinations:

$$\sum_{j=1}^{t}\gamma_j\psi(v_j)r_j,$$

for some $t \in \mathbb{N}$, where simplex $\gamma \in \Delta^t, \|r_j\| \leqslant \frac{B_X}{\lambda}, \forall j \in [t]$. We have $\widehat{\mathcal{U}}$ in the following form:

$$\widehat{\mathcal{U}} = \Psi Q = \sum_{i=1}^{n}\psi(y_i)q_i,$$

where $\|q_i\| \leqslant \frac{B_X}{n\lambda}, \forall i \in [n]$. Now we set $q_i = \gamma_i r_i$, where $\gamma_i = 1/n$ satisfies $\gamma \in \Delta^n$, and for each $i$, we have:

$$\|q_i\| = \frac{1}{n} \cdot \|r_i\| \leqslant \frac{B_X}{n\lambda} \implies \|r_i\| \leqslant \frac{B_X}{\lambda}.$$

Therefore, we can rewrite $\widehat{\mathcal{U}}$ as:

$$\widehat{\mathcal{U}} = \sum_{i=1}^{n}\frac{1}{n} \cdot \psi(y_i)r_i,$$

where each $\psi(y_i)r_i$ is the atom of $\mathcal{D}$, which means $\widehat{\mathcal{U}} \in \mathcal{D}$. $\qquad\square$

**Lemma C.2.** *Suppose $\mathcal{D}$ is the convex hull of set $S$, defined in Equation* (17), *and denote $d_{\mathcal{D}}$ and $d_S$ are the diameters of $\mathcal{D}$ and $S$ respectively, then $d_{\mathcal{D}} = d_S = \frac{2B_X B_Y}{\lambda}$.*

*Proof.* We first denote $s \in \mathcal{D}_0$ as the atom of $\mathcal{D}$, i.e., $s = \psi(v)r$. On the one hand, $\mathcal{D}$ contains all elements expressible as finite linear combinations, i.e., $U = \sum_{j=1}^{t} \gamma_j s_j$, where $t \in \mathbb{N}, \gamma \in \Delta^t$. According to the triangle inequality and the convexity of $\mathcal{D}$, $\forall U, U' \in \mathcal{D}$, we have:

$$\|U - U'\| \leqslant \max_{i,j} \left\| s_i - s'_j \right\| \leqslant d_{\mathcal{D}_0},$$

which means:

$$d_{\mathcal{D}} = \sup_{U,U' \in \mathcal{D}} \|U - U'\| \leqslant \sup_{s_i, s'_j \in S} \left\| s_i - s'_j \right\| = d_{\mathcal{D}_0}.$$

On the other hand, $\mathcal{D} = \mathrm{conv}(\mathcal{D}_0)$ means $\mathcal{D}_0 \subseteq \mathcal{D}$, then $d_{\mathcal{D}_0} \leqslant d_{\mathcal{D}}$. Therefore:

$$d_{\mathcal{D}} = d_{\mathcal{D}_0} = \sup_{s,s' \in \mathcal{D}_0} \|s - s'\| = \sup_{s \in \mathcal{D}_0} 2\|s\| = \sup_{s = \psi(v)r} 2\|\psi(v)\| \cdot \|r\| = \frac{2B_X B_Y}{\lambda}.$$

$\square$

**Lemma C.3.** *The curvature constant of function $f$ defined in Equation* (18) *is $C_f = \left(\frac{2B_X B_Y}{\lambda}\right)^2$.*

*Proof.* The curvature constant of a function $f$ is defined as:

$$C_f = \sup_{\substack{U,U' \in \mathcal{D}, \alpha \in [0,1] \\ y = U + \alpha(U' - U)}} \frac{2}{\alpha^2} \Big( f(y) - f(U) - \langle \nabla f(U), y - U \rangle \Big).$$

The Hessian matrix of $f$ is $\nabla^2 f(U) = I$, therefore, according to the second-order Taylor expansion, we have:

$$\begin{aligned}
f(y) &= f(U) + \langle \nabla f(U), y - U \rangle + \frac{1}{2} \left\langle y - U, \nabla^2 f(U)(y - U) \right\rangle \\
&= f(U) + \langle \nabla f(U), y - U \rangle + \frac{1}{2} \|y - U\|^2 \\
&= f(U) + \langle \nabla f(U), y - U \rangle + \frac{\alpha^2}{2} \|U' - U\|^2.
\end{aligned}$$

Substituting this into the curvature constant definition, and according to Lemma C.2, we have:

$$\begin{aligned}
C_f &= \sup_{\substack{U,U' \in \mathcal{D} \\ \alpha \in [0,1]}} \frac{2}{\alpha^2} \cdot \frac{\alpha^2}{2} \|U' - U\|^2 \\
&= \sup_{U,U' \in \mathcal{D}} \|U' - U\|^2 = d_{\mathcal{D}}^2 = \left(\frac{2B_X B_Y}{\lambda}\right)^2.
\end{aligned}$$

$\square$

**Lemma C.4.** *For a step $U_{t+1} = U_t + \alpha_t(\psi(v_{t+1})r_{t+1} - U_t)$ with $\alpha_t = 1/(t+1)$, it holds that:*

$$f(U_{t+1}) \leqslant f(U_t) - \alpha_t g(U_t) + \frac{\alpha_t^2}{2} C_f,$$

*where $g(U) = \max_{U' \in \mathcal{D}} \langle \nabla f(U), U - U' \rangle$ is the current duality gap, and $C_f$ is the curvature constant of $f$.*

*Proof.* Lemma C.4 is a special case of lemma 5 in Jaggi (2013), Appendix A. $\square$

## C.5. The Proof of the Upper Bound Theorem

*Proof.* We want to prove Theorem 3.4. Due to the convexity of function $f$, we have $\forall U \in \mathcal{D}$:

$$f(U) - f(U^*) \leqslant \langle \nabla f(U), U - U^* \rangle,$$

where $U^* = \widehat{\mathcal{U}}$. Then:

$$\begin{aligned} g(U) &= \max_{U' \in \mathcal{D}} \langle \nabla f(U), U - U' \rangle \\ &\geqslant \langle \nabla f(U), U - U^* \rangle \\ &\geqslant f(U) - f(U^*). \end{aligned}$$

According to Lemma C.1, we have $f(U^*) = 0$, thus $g(U) \geqslant f(U)$. Then according to Lemma C.4, we have:

$$\begin{aligned} f(U_{t+1}) &\leqslant f(U_t) - \alpha_t g(U_t) + \frac{\alpha_t^2}{2} C_f \\ &\leqslant f(U_t) - \alpha_t f(U_t) + \frac{\alpha_t^2}{2} C_f \\ &\leqslant (1 - \alpha_t) f(U_t) + \frac{\alpha_t^2}{2} C_f. \end{aligned}$$

Substituting the step size $\alpha_t = 1/(t+1)$:

$$\begin{aligned} f(U_{t+1}) &\leqslant \frac{t}{t+1} f(U_t) + \frac{C_f}{2(t+1)^2} \\ \implies (t+1) f(U_{t+1}) &\leqslant t f(U_t) + \frac{C_f}{2(t+1)}. \end{aligned}$$

Letting $m = t + 1$ and applying induction:

$$\begin{aligned} m f(U_m) &\leqslant (m-1) f(U_{m-1}) + \frac{C_f}{2m} \\ &\leqslant \frac{C_f}{2} \sum_{i=1}^{m} \frac{1}{i} \leqslant \frac{C_f(\ln m + 1)}{2}. \end{aligned}$$

From Lemmas C.2 and C.3, the curvature constant is $C_f = d_{\mathcal{D}}^2 = \left( \frac{2B_X B_Y}{\lambda} \right)^2$. Therefore:

$$\begin{aligned} f(U_m) = \frac{1}{2} \left\| U_m - \widehat{\mathcal{U}} \right\|_{\mathrm{HS}}^2 &\leqslant \frac{\left( \frac{2B_X B_Y}{\lambda} \right)^2 (\ln m + 1)}{2m} \\ \implies \left\| U_m - \widehat{\mathcal{U}} \right\|_{\mathrm{HS}} &\leqslant \frac{2B_X B_Y}{\lambda} \sqrt{\frac{\ln m + 1}{m}}. \end{aligned}$$

According to Song et al. (2009), the empirical CME estimator $\widehat{\mathcal{U}}$ converges to the true CME $\mathcal{U}$ in the RKHS norm in probability at the rate $O_p \left( \frac{1}{\sqrt{n\lambda}} + \sqrt{\lambda} \right)$. Since $\widetilde{\mathcal{U}} = U_m$, we have:

$$\begin{aligned} \left\| \mathcal{U} - \widetilde{\mathcal{U}} \right\|_{\mathrm{HS}} &= \left\| \mathcal{U} - \widehat{\mathcal{U}} + \widehat{\mathcal{U}} - \widetilde{\mathcal{U}} \right\|_{\mathrm{HS}} \\ &\leqslant \left\| \mathcal{U} - \widehat{\mathcal{U}} \right\|_{\mathrm{HS}} + \left\| \widehat{\mathcal{U}} - \widetilde{\mathcal{U}} \right\|_{\mathrm{HS}} \\ &= O_p \left( \frac{1}{\sqrt{n\lambda}} + \sqrt{\lambda} + \frac{1}{\lambda} \sqrt{\frac{\ln m}{m}} \right). \end{aligned}$$

$\square$

## D. The Pseudocode of RNCME Learnware Recommendation

In this section, we provide the pseudocode of the RNCME learnware recommendation algorithm, which can be divided into four parts: the neural feature map training (Algorithm 1), the reduced set generation (Algorithm 2), the learnware submission (Algorithm 3) and the learnware deployment (Algorithm 4).

### D.1. The Pseudocode of Training Neural Feature Map

---

**Algorithm 1** Training Neural Feature Map $\phi_\theta$

---

**Require:** The i.i.d. dataset $(X, Y) \sim P_{XY}$, the output kernel $k_{\mathcal{Y}}$, the regularization parameter $\lambda$, batch size $b$, step size $\eta_\theta$
**Ensure:** The well-trained neural feature map $\phi_\theta$
1: Initialize $\phi_\theta$ randomly
2: **while** not converged **do**
3: $\quad (X_b, Y_b) \sim \text{Mini-batch}(X, Y)$
4: $\quad K_Y \leftarrow k_{\mathcal{Y}}(Y_b, Y_b) \in \mathbb{R}^{b \times b}$
5: $\quad \Phi_\theta \leftarrow \phi_\theta(X_b) \in \mathbb{R}^{d \times b}$
6: $\quad \mathcal{L}(\theta) \leftarrow -\text{tr}\left( K_Y \Phi_\theta^\mathrm{T} \left( \Phi_\theta \Phi_\theta^\mathrm{T} + b\lambda I \right)^{-1} \Phi_\theta \right)$
7: $\quad \theta \leftarrow \theta - \eta_\theta \nabla_\theta \mathcal{L}(\theta)$
8: **end while**
9: **return** $\phi_\theta$

---

### D.2. The Pseudocode of the Reduced Set Generation

---

**Algorithm 2** Reduced Set Generation

---

**Require:** The i.i.d. dataset $(X, Y) \sim P_{XY}$, the output kernel $k_{\mathcal{Y}}$, the neural feature map $\phi_\theta$, size of reduced set $m$, regularization parameter $\lambda$, bound $B_X$
**Ensure:** The reduced set $V = \{v_j\}_{j=1}^m$ and the weight matrix $R \in \mathbb{R}^{m \times d}$
1: Compute feature matrix $\Phi_\theta \leftarrow \left( \phi_\theta(x_1) \quad \cdots \quad \phi_\theta(x_n) \right) \in \mathbb{R}^{d \times n}$.
2: Compute weight matrix $Q \leftarrow \Phi_\theta^\mathrm{T}(\Phi_\theta \Phi_\theta^\mathrm{T} + n\lambda I)^{-1} \in \mathbb{R}^{n \times d}$.
3: Initialize $V \leftarrow \varnothing, R \leftarrow \varnothing$.
4: **for** $t = 0 \rightarrow m - 1$ **do**
5: $\quad v_{t+1} \leftarrow \arg\max_{v \in \mathcal{Y}} \|h_t(v)\|^2$
6: $\quad r_{t+1} \leftarrow \dfrac{B_X}{\lambda} \cdot \dfrac{h_t(v_{t+1})}{\|h_t(v_{t+1})\|}$.
7: $\quad V \leftarrow V \cup \{v_{t+1}\}, R \leftarrow \begin{pmatrix} R \\ r_{t+1} \end{pmatrix}$.
8: **end for**
9: **return** $V$ and $R/m$.

---

### D.3. The Pseudocode of Learnware Submission and Deployment

---

**Algorithm 3** Learnware Submission

---

**Require:** The i.i.d. dataset $(X, Y) \sim P_{XY}$, the machine learning model $f$ well-trained on $(X, Y)$, the output kernel $k_{\mathcal{Y}}$
1: Train $\phi_\theta$ via Algorithm 1 based on $(X, Y)$
2: Generate $V$ and $R$ via Algorithm 2 based on $(X, Y)$, $\phi_\theta$ and $k_{\mathcal{Y}}$
3: Submit $(f, V, R)$ to the learnware market

---

---

**Algorithm 4** Learnware Deployment

---

**Require:** The small i.i.d. labeled dataset $(X_l, Y_l) \sim P_{XY}$, the large i.i.d. unlabeled dataset $X_u \sim P_X$, the learnware
market $\{(f_c, V_c, R_c)\}_{c=1}^{N}$, the output kernel $k_{\mathcal{Y}}$

**Ensure:** The recommended learnware $f_{c^*}$

1: Train $\phi_\theta$ via Algorithm 1 based on $(X_l, Y_l)$
2: Generate $V_t$ and $R_t$ via Algorithm 2 based on $(X_l, Y_l)$, $\phi_\theta$ and $k_{\mathcal{Y}}$
3: Submit $(V_t, R_t)$ to the learnware market
4: Receive $f_{c^*}$ recommended by the market, where $c^* = \arg\min_{1 \leqslant c \leqslant N} \left\| \widetilde{\mathcal{U}}_c - \widetilde{\mathcal{U}}_t \right\|_{\mathrm{HS}}^2$, and:

$$
\begin{aligned}
\left\| \widetilde{\mathcal{U}}_c - \widetilde{\mathcal{U}}_t \right\|_{\mathrm{HS}}^2 &= \| \Psi_{V_c} R_c - \Psi_{V_t} R_t \|_{\mathrm{HS}}^2 \\
&= \mathrm{tr}\left( R_c^{\mathrm{T}} K_{V_{cc}} R_c \right) - 2\mathrm{tr}\left( R_c^{\mathrm{T}} K_{V_{ct}} R_t \right) + \mathrm{tr}\left( R_t^{\mathrm{T}} K_{V_{tt}} R_t \right) \\
&\propto \mathrm{tr}\left( R_c^{\mathrm{T}} K_{V_{cc}} R_c \right) - 2\mathrm{tr}\left( R_c^{\mathrm{T}} K_{V_{ct}} R_t \right).
\end{aligned}
$$

5: Deploy $f_{c^*}$ for the unlabeled dataset $X_u$.

---

# E. Experimental Details

Due to the space limitation, we provide more experimental details in this appendix.

## E.1. Hardware and Software Configuration

All experiments are conducted on an Ubuntu 20.04 LTS server equipped with an AMD EPYC 7H12 64-core processor (256 threads), 1TB RAM, and an NVIDIA A100 PCIe GPU (80GB VRAM). The code is implemented in Python 3.12.7.

## E.2. Real-World Regression Experiments

We conduct regression experiments on California Housing Prices dataset (Nugent, 2017), which contains 20,640 samples with 10 features: longitude, latitude, housing median age, total rooms, total bedrooms, population, households, median income, median house value, and ocean proximity. We convert the categorical ocean proximity feature into numerical values by mapping "NEAR BAY" to 0, "INLAND" to 1 and "<1H OCEAN" to 2. All features are then normalized to $[0, 1]$.

We create 42 tasks for constructing the learnware market and 21 tasks for simulating user scenarios. For each learnware task, we train a Gradient Boosted Decision Tree (GBDT) (Friedman, 2001) model with 100 estimators, a maximum depth of 3, and a learning rate of 0.1. We compare our RNCME method with the basic RKME method (Wu et al., 2023), as all labeled-RKME improvements are constrained to classification tasks and thus not applicable here.

For RKME, we set the reduced set size to $m = 1{,}500$ and perform 5 update steps with a learning rate of 0.1 (we note that $m = 50$ is already sufficient for convergence; the larger $m = 1{,}500$ is set for a fair comparison). For RNCME, the neural feature map is implemented as a 16-dimensional linear layer followed by a Sigmoid activation function (to bound the output). The output kernel $k_{\mathcal{Y}}$ is set as the Gaussian kernel with bandwidth $1/d_Y$, where $d_Y$ is the dimension of the output space, and $\psi$ is the implicit feature map. Training is performed for 2 epochs with a learning rate of $10^{-4}$ using the entire dataset. Since capturing conditional distributions requires more samples than capturing marginal distributions, we set a larger $m = 1{,}500$ for RNCME (compared with $m = 50$). To solve Equation (7), we initialize $V$ via $k$-means clustering and then optimize it using gradient ascent for 5 steps with a learning rate of 0.1. The regularization parameter $\lambda$ is set to $10^{-3}$.

## E.3. Real-World Classification Experiments

We conduct classification experiments on NICO$^{++}$ dataset (Zhang et al., 2023), which is designed for OOD image classification. We create 30 tasks for constructing learnware market and 20 tasks for simulating user scenarios. For each learnware task, we train a neural network with a frozen DenseNet201 (Huang et al., 2017) backbone and a trainable 2-layer MLP classifier. The backbone is initialized with ImageNet-pretrained weights, outputting 1920-dimensional features. The MLP classifier receives feature vectors from the backbone network as input. It processes these features through a two-layer architecture: a hidden layer of dimension 256 with ReLU activation and dropout ($p = 0.5$), followed by an output layer

that produces the final predictions. We train the model for 100 epochs using Adam optimizer with a learning rate of $10^{-3}$ and a batch size of 32,768. We compare our RNCME method with the basic RKME method (Wu et al., 2023) and all the labeled-RKME improvements, including LP (Guo et al., 2023), HF (Tan et al., 2024a) and cRKME (Liu et al., 2025).

For RKME, we set the reduced set size to $m = 1,300$ ($m = 100$ is enough) and perform 5 update steps with a learning rate of 0.1. For RNCME, the neural feature map is implemented as a 64-dimensional linear layer followed by a Sigmoid activation function (to bound the output). The output kernel $k_{\mathcal{Y}}$ is set as the Gaussian kernel with bandwidth $1/d_Y$, where $d_Y$ is the dimension of the output space, and $\psi$ is the implicit feature map. Training runs for 5 epochs with a learning rate of $10^{-4}$ using the entire dataset. We use a reduced set size of $m = 1300$ for RNCME. To solve Equation (7), we search over the discrete label space $\mathcal{Y}$ to find the optimal $v$. The regularization parameter $\lambda$ is set to 0.1. For LP, we distill the linear proxy 100 for steps with a learning rate of 0.01. For HF, we set the reduced set size to $m = 1,300$ ($m = 200$ is enough) and perform 5 update steps with a learning rate of 0.1. For cRKME, we use the same configuration as HF, with an additional similarity regularization parameter of 0.1.

### E.4. Time Efficiency Analysis

All experimental configurations remain the same as in Section E.3, except that for the RNCME regression experiment, we use the NICO$^{++}$ dataset and use the enumurical value of the category as the regression label, and we solve the discrete optimization problem in Equation (7) via gradient ascent instead of searching over the discrete label space $\mathcal{Y}$, as we are only measuring the time for solving the reduced set, independent of the task itself.

### E.5. The Reduced Set Generation

All experimental configurations remain the same as in Section E.3.

### E.6. The Effectiveness of Training Neural Feature Map

We construct a hand-crafted, highly complex conditional distribution $P_{Y|X}$, where $X \in \mathbb{R}^{256}$ and $Y \in \mathbb{R}^{128}$. First, we sample $X$ from a uniform distribution over $[-3, 3]^{256}$, and then we generate $Y$ based on different "region" defined by the norm $r = \|x_{1:8}\|$ of the first 8 dimensions of $X$:

- **Region 1** ($r < 1.0$)**:** A multimodal distribution. With probability 0.7, $Y$ is generated through $\tanh(W_1 x_{1:8}) + \epsilon_1$, where $W_1 \sim \mathcal{N}(0, 0.1)$; otherwise through $\sin(W_2 x_{9:16}) + \epsilon_2$, where $W_2 \sim \mathcal{N}(0, 0.15)$.

- **Region 2** ($1.0 \leq r < 2.0$)**:** A heteroscedastic Gaussian $Y \sim \mathcal{N}(\mu(x), \sigma^2(x))$, where $\mu(x) = W_\mu x_{1:16}$ with $W_\mu$ scaled by sinusoidal functions, and $\sigma(x) = 0.1 + 0.4 \cdot \text{sigmoid}(\bar{\theta})$, where $\bar{\theta}$ is the mean of $x_{1:16}$.

- **Region 3** ($r \geq 2.0$)**:** A three-component mixture. Component A applies sparse masking to a Gaussian; Component B uses low-rank linear mapping; Component C generates piecewise-constant outputs via $\text{sign}(x)$.

All outputs are perturbed by isotropic Gaussian noise $\epsilon \sim \mathcal{N}(0, 0.05^2)$. This design incorporates multiple challenges: multimodality, heteroscedasticity, non-linear mappings, and region-dependent behaviors, making it a rigorous testbed for conditional distribution estimation.

On the constructed dataset, we investigate the impact of training the neural feature map $\phi_\theta$ on CME estimation quality. Here $\phi_\theta$ is implemented as a 64-dimensional linear layer followed by a Sigmoid activation function (to bound the output), and it is trained for 40 epochs with a learning rate of $5 \times 10^{-4}$. The regularization parameter $\lambda$ is set to 0.01. And for the RBF kernel CME, $\lambda$ is set to 0.001.

