# OpenReview forum: "Identifying Learnwares via Reduced Neural Conditional Mean Embedding"
_ICML.cc/2026/Conference — ICML 2026 regular_

### Official Review · Reviewer_TZyR · 2026-03-02

**Soundness:** 3
**Presentation:** 2
**Significance:** 3
**Originality:** 3
**Overall Recommendation:** 3
**Confidence:** 5

**Summary:**

This paper proposes a novel learnware paradigm specification, Reduced Neural Conditional Mean Embedding (RNCME), addressing the limitations of existing RKME specifications that utilize label information—namely, the lack of a theoretical explanation and restrictions on classification tasks.

**Compliance With Llm Reviewing Policy:**

Affirmed.

**Key Questions For Authors:**

1. What are the specific settings for $\psi$ in the experiment? Are there any special encoding operations in the label space?
2. Regarding the HF work by Tan et al. (2024a), the paper states that it cannot handle regression tasks, and your argument suggests that all labeled-RKME-related work is incapable of handling regression tasks. However, I found in the HF paper that the method can indeed handle regression tasks in their experiments.
3. Regarding Section 3.3, is there any research paper or experimental evidence supporting the claim that trained neural feature maps alleviate the absence of the characteristic property? Furthermore, does the quality of the mapping training affect the performance of subsequent specification generation? Additionally, in the learnware paradigm, the labels of input data are typically the ground-true labels of the data, differing from the settings described in the paper.
4. Does the RNCME specification yield a label reduction set V and weight matrix R capable of handling mixed-task scenarios present in the learnware paradigm?
5. What is the role of the unlabeled dataset input in the Algorithm 4?

**Limitations:**

yes

**Strengths And Weaknesses:**

This paper extends labeled-RKME to classification and regression tasks from a theoretical perspective based on the CME. It then proposes a data-adaptive neural feature map that addresses the kernel selection dependency and high computational complexity issues inherent in previous CME approaches. Furthermore, by employing an algorithm for solving constrained vector-weighted low-rank kernel approximation problems, it generates a reduced set approximating empirical CME to construct the RNCME specification method. Finally, a learnware paradigm is constructed using this specification method. The methodology, theory, and experiments in this paper effectively validate the proposed approach and demonstrate that it resolves the limitations described in the related work. However, the writing and viewpoints of this paper require further examination, revision, and careful consideration. For instance, the theory mentioned as Theorem 3.1, 3.2, 3.3 does not appear in the paper.

---

> ### Author Rebuttal · Authors · 2026-03-26
>
> We thank the reviewers for their time and constructive feedback. Below we address the comments point by point.
>
> ### Weakness: On the writing and viewpoints concern (e.g., the missing theorems)
>
> We sincerely apologize for this oversight. This was caused by a misconfiguration of the \cref command in LaTeX. We mistakenly referenced "Theorem" while the actual environments were "Assumption", "Corollary", and "Proposition". The numbering itself remains consistent and unique (Assumption 3.1, Corollary 3.2, Proposition 3.3), so the intended references are still clear. We are sorry for the confusion.
>
> ### Question 1: On the specific settings for $\psi$ in the experiment
>
> As mentioned at the end of the first paragraph in Section 4, we placed the detailed experimental setup in Appendix E due to space limitations. As described in Appendices E.2 and E.3, the output kernel $k_\mathcal{Y}$ is a Gaussian kernel, and $\psi$ is its implicit feature map, which is ultimately represented as a kernel matrix.
>
> ### Question 2: On whether HF (Tan et al. 2024a) supports regression
>
> HF primarily addresses the problem of "heterogeneous feature spaces", which has been the focus of their prior work. Incorporating output information serves as an improvement to their existing framework. Their method consists of two main components:
>
> 1. RKME_L: a labeled-RKME improvement, which is the same purpose as ours (i.e., generating specifications that capture input-output relationships). As shown in their Appendix E.3, Figure 6, RKME_L doesn't support regression.
> 2. Subspace learning: a component that handles heterogeneous feature spaces, building on their prior work. While it can be applied to regression tasks, it addresses a different problem.
>
> We include HF as a baseline because RKME_L itself can serve as a standalone specification method. Even without the subsequent subspace learning step, using RKME_L allows learnware recommendation to benefit from capturing conditional distributions. Therefore, it is a natural and fair baseline. We acknowledge that this distinction was not made sufficiently clear in our paper, and we sincerely apologize for the confusion this caused.
>
> ### Question 3
>
> #### On the neural feature maps
>
> To the best of our knowledge, there is currently no theoretical guarantee that a finite-dimensional neural feature map recovers the characteristic property. We leverage the strong representational capacity of neural networks to alleviate its absence. This perspective is consistent with Shimizu et al. (2024), who also adopt a neural feature map to address the expressiveness limitations of kernel-based CMEs without relying on characteristic properties.
>
> We provide empirical evidence supporting the effectiveness of our approach in Section 4.5, where we varied the number of training epochs. The results demonstrate that proper training of $\phi_\theta$ is essential for achieving high-quality CME estimation, which in turn affects the subsequent specification generation and matching performance.
>
> #### On the use of ground-truth labels
>
> We confirm that the labels used in our experiments are indeed the ground-truth labels of the data. In the submitting stage, developers train $\phi_{\theta_c}$ and generate specifications using their local datasets $(X_c,Y_c)$ with ground-truth labels. In the deploying stage, users similarly use their small labeled dataset $(X_l, Y_l)$ with ground-truth labels to train $\phi_{\theta_t}$ and generate the task specification. This is consistent with the standard learnware setup.
>
> ### Question 4: On whether RNCME can handle mixed-task scenarios
>
> In the current learnware paradigm, classification and regression tasks are typically not mixed during recommendation: the market allows users to specify the task type, and retrieval is performed within the corresponding category. Even the output-ignoring RKME would not recommend a regression model to a user with a classification task. Therefore, the fact that RNCME does not explicitly handle cross-type mixing is not a limitation specific to our method.
>
> ### Question 5: On the role of unlabeled data in Algorithm 4
>
> We thank the reviewer for pointing out this omission. In the deployment stage, the user's large unlabeled dataset $D_u$ is the target data on which the recommended learnware $f_{c^∗}$ will be applied. We will add an explicit step in Algorithm 4 to indicate that the returned learnware is used to predict on $D_u$. We apologize for this omission.
>
> We hope these clarifications and revisions fully address your concerns.
>
> ### References
>
> - Tan, P., Liu, H.-T., Tan, Z.-H., and Zhou, Z.-H. Handling learnwares from heterogeneous feature spaces with explicit label exploitation. In Advances in Neural Information Processing Systems, volume 37, pp. 12767–12795, 2024a.
> - Shimizu, E., Fukumizu, K., and Sejdinovic, D. Neural kernel conditional mean embeddings. In International Conference on Machine Learning, 2024.

---

> > ### Author Rebuttal · Reviewer_TZyR · 2026-04-03
> >
> > The author mentions a number of appendices in his response, but I didn't find them. Moreover, a mixed task scenario refers to a scenario where the task exceeds the capabilities of a single model.

---

> > > ### Author Response · Authors · 2026-04-04
> > >
> > > Thank you for your acknowledgement. However, we respectfully point out two factual issues in your statement.
> > >
> > > ## Appendices
> > >
> > > We apologize for the confusion. Appendix E.2 and E.3 are on **Page 18** of our PDF, where we detail the experimental settings for regression and classification tasks. Specifically, $k_{\mathcal{Y}}$ is a Gaussian kernel and $\psi$ is its implicit feature map, represented as a kernel matrix.
> > >
> > > ## On mixed-task scenarios
> > >
> > > First, we thank the reviewer for clarifying the definition of "mixed-task scenario". In our paper, we repeatedly mention that task types are divided into classification and regression. Therefore, when we first saw the term "mixed-task", our immediate reaction was mixing between classification and regression. We apologize for this misunderstanding and thank the reviewer for their patient clarification.
> > >
> > > The mixed-task scenario defined by the reviewer, which refers to a scenario where the task exceeds the capabilities of a single model, is actually a broad problem. Solving this problem typically requires multiple steps, such as identifying suitable models, combining multiple models, or adapting models to new tasks.
> > >
> > > Our work focuses on the **first and most fundamental step** of this broad problem: **identifying the model that best matches the user's task from the market**. RNCME achieves this by comparing input-output conditional distributions, rather than comparing specific inputs or specific outputs. As illustrated in the third paragraph of our Introduction, consider a user with a "Cartoon Character Emotion Prediction" task and two candidate learnwares: one trained on cartoon faces for character recognition, and the other trained on human faces for emotion prediction. Although the input styles differ, the underlying logic of identifying expressions (e.g., the curve of a mouth or the narrowing of eyes) is shared between the user's task and the second model. RNCME can identify the second model as more suitable. This itself is a scenario where the task exceeds the capability of a single model, and RNCME can effectively identify the most relevant model.
> > >
> > > As for the other steps of this broad problem, such as identifying and combining multiple models to solve complex tasks, existing research has already addressed them. For example, Zhang et al. (2021) studied how to adapt models in the market to unseen user tasks based on Mixture Proportion Estimation with RKME. These approaches rely on underlying specifications such as RKME. Our RNCME can directly replace RKME and thus be seamlessly integrated into these methods.
> > >
> > > Therefore, we believe that RNCME provides an important building block for solving mixed-task problems. We focus on the core step of identifying the most relevant model, while subsequent steps of combination and adaptation can be built on top of our work.
> > >
> > > ## Reference
> > >
> > > Zhang, Y.-J., Yan, Y.-H., Zhao, P., and Zhou, Z.-H. Towards enabling learnware to handle unseen jobs. In Proceedings of the AAAI Conference on Artificial Intelligence, volume 35, 2021.

---

### Official Review · Reviewer_DGBd · 2026-03-17

**Soundness:** 4
**Presentation:** 4
**Significance:** 3
**Originality:** 3
**Overall Recommendation:** 5
**Confidence:** 3

**Summary:**

This paper proposes RNCME, a new specification generation method for the learnware paradigm. The key observation is that existing RKME specifications only capture the marginal input distribution, causing functionally different models trained on similar inputs to receive similar specifications. RNCME addresses this by directly modeling $P_{Y|X}$ via CME. A data-adaptive neural feature map is introduced to reduce the $O(n^3)$ computational cost of standard CME to $O(nd^2 + d^3)$, and a Frank-Wolfe-based iterative algorithm generates a privacy-preserving reduced set approximating the empirical CME. A theoretical error bound is provided. The method handles both regression and classification, unlike prior RKME improvements which are restricted to classification.

**Compliance With Llm Reviewing Policy:**

Affirmed.

**Key Questions For Authors:**

The deployment stage requires training $\phi_\theta$ on the user's small labeled dataset $D_l$. Has the sensitivity of recommendation quality to the size of $|D_l|$ been evaluated? This seems important for validating practical applicability.

**Limitations:**

The paper does not explicitly discuss limitations. The authors should note that the framework assumes a shared output space across learnwares and user tasks, and that behavior under very limited user-side labeled data has not been characterized.

**Strengths And Weaknesses:**

Soundness. The paper is technically solid. The error decomposition into statistical and reduction components is well-grounded, and the bound in Theorem 3.4 is a strength relative to prior works, which lack comparable theoretical analysis. The ablation study on the neural feature map is well-designed. One gap is the absence of any analysis on how performance varies with the amount of user-side labeled data, which is central to the deployment assumptions of the learnware paradigm.
Presentation. The paper is clearly written and the motivation is easy to follow. The narrative from problem identification to method design is coherent. The connection between CME estimation and function-valued ridge regression could be explained more accessibly, but this is a minor issue.
Significance. Extending learnware specifications to regression is a real and non-trivial contribution, since existing RKME approaches are structurally limited to classification. The computational efficiency gains are practically meaningful and well-evidenced by the experiments.
Originality. The individual components are not new, but their integration within the learnware framework is well-motivated and non-trivial. The paper clearly distinguishes its contributions from prior work.

---

> ### Author Rebuttal · Authors · 2026-03-26
>
> We thank the reviewers for their time and constructive feedback. Below we address the comments point by point.
>
> ### Weakness & Question: On the sensitivity to user-side labeled data size $|D_l|$
>
> We acknowledge that in extremely few-shot settings, such as fewer than 5 labeled samples, our method like other labeled-RKME improvements may struggle due to insufficient data. Empirically, we observe that with only a few dozen labeled samples, RNCME already outperforms the original RKME. The table below shows the NDCG@1 values for RNCME as $|D_l|$ varies, with RKME as the baseline.
>
> | Method  | RKME (baseline) | RNCME | RNCME | RNCME | RNCME | RNCME |
> | ------- | --------------- | ----- | ----- | ----- | ----- | ----- |
> | $\mid D_l\mid $ | —               | 1     | 5     | 10    | 20    | 50    |
> | NDCG@1  | 0.774           | 0.421 | 0.710 | 0.786 | 0.799 | 0.810 |
>
> For RKME, the reduced size $m=50$, and $|D_l|$ is not applicable since no labeled data used.
>
> ### Limitation: On the shared output space assumption
>
> In our experiments (Section 4.2), we randomly sampled 30~40 classes from a total of 80 to construct each task, ensuring that no two tasks shared exactly the same set of output categories. This simulates a scenario with partially overlapping but not identical output spaces. However, we acknowledge that more heterogeneous output spaces (e.g., mixing digital labels and text) remain an important challenge. Extending RNCME to handle such cases is an interesting direction for future work, and we will add a discussion of this limitation in the final version.
>
> We thank the reviewer again for their valuable feedback, which will help us strengthen the paper.

---

### Official Review · Reviewer_B1jG · 2026-03-22

**Soundness:** 3
**Presentation:** 3
**Significance:** 3
**Originality:** 3
**Overall Recommendation:** 5
**Confidence:** 3

**Summary:**

- This paper studies the task generating learnware specifications using the conditional mean embedding
- This builds on prior work which use the standard mean embeddings/reduced versions (RKME), and highlights the issue that these models only model the input distribution, but not the input-output distribution relationship
- To address this, the authors propose using Conditional Mean Kernel Embeddings, but note CMEs are expensive to compute, requiring inverting and NxN matrix (N = # of data points)
- To deal with the compute issues, they use a finite dimensional kernel for in the input kernel, which is parameterized b a neural network trained to make the resulting CME as accurate as possible on the regression task proposed in equation 3
- Next, to reduce the number of released data points, they proposed using a reduced set. Optimizing this reduced set is done via a Frank-Wolfe algorithm
- Empirically, the method strongly outperforms prior methods. The authors also perform experiments showing the effectiveness of the neural kernel and also the runtime of the algorithm

**Compliance With Llm Reviewing Policy:**

Affirmed.

**Final Justification:**

The reviewers clarified my concerns. I don't think any issues were raised during the review period that would cause me to change my score.

**Key Questions For Authors:**

- Can the authors elaborate more of the requirement of developers to train their own ϕ_θ? Why does it make sense to directly compare U_c and U_t in equation 12, given that one is computed using the learnware provider's ϕ_θ, and the other is computed using the developers?
- Can the authors provide more clarity on the choice of metric in equation 12? Even if ϕ_θ is identical, wouldn't it make more sense to use the developer's distribution on it? I.e. do $E_{x \sim P_X^t} \| U_c \phi(x) - U_t \phi(x) \|_G^2$

**Limitations:**

See weaknesses/questions.

**Strengths And Weaknesses:**

Strengths
- Well motivated, and a clear and direct application of the CME for the learnware task
- Issues with using the CME directly are explained clearly and systematically resolved in a principled manner(eg. the inversion cost -> finite dimensional kernels -> neural approximations)
- Some theory is provided for the subset selection algorithm
- Empirical perform is strong and ablations clearly show the importance of the design choices

Weaknesses
- There isn't much justification for the choice of the learnware matching metric, c* (equation 12)
- This method requires the developer to locally train their own feature map ϕ_θ. I think this limitation should be discussed more in depth, as this additional training step also adds to the compute cost of the method, and is not present in prior work

---

> ### Author Rebuttal · Authors · 2026-03-26
>
> We thank the reviewer for their positive assessment. We address the two questions below.
>
> ### Question 1 (Weakness 2): On the training of $\phi_{\theta}$
>
> We acknowledge that training $\phi_{\theta}$ introduces an additional step not present in the original RKME method. However, we note that other labeled-RKME improvements also involve extra computations. For example, LP requires Hungarian algorithm matching, and cRKME involves max-flow min-cost optimization. Moreover, the additional cost of training $\phi_\theta$ is modest in practice. As shown in Table 1, the Compression column for RNCME includes both training $\phi_\theta$ and generating the reduced set. For classification tasks, this process takes only 0.516 seconds, and for regression tasks, it takes 26.65 seconds (with the difference primarily due to the reduced set generation step), which remains within an acceptable range compared to other labeled-RKME improvements.
>
> Additionally, $\widetilde{\mathcal{U}}_c$ and $\widetilde{\mathcal{U}}_t$ remain comparable despite different $\phi\_{\theta\_c}$ and $\phi\_{\theta\_t}$, since both $\mathcal{H}\_{\theta\_c}$ and $\mathcal{H}\_{\theta\_t}$ are subspaces of $\mathbb{R}^d$, endowing them with a common geometric structure that enables consistent similarity measurement.
>
> ### Question 2 (Weakness 1): On the matching metric in Equation (12)
>
> The reviewer raises a valid point. The metric in equation 12 directly compares the conditional mean embedding operators, which encode the full input-output relationship. A smaller HS norm between two CMEs implies more similar conditional distributions, which is precisely what we want for functional matching.
>
> The alternative metric suggested by the reviewer is indeed a natural way to measure how well the candidate learnware performs on the user's input distribution. However, this would require the user to reveal their input distribution $P_X^t$, which may raise privacy concerns. In the learnware paradigm, the user submits only the specification $\widetilde{\mathcal{U}}_t$ (which is privacy-preserving) rather than raw data. Our metric allows matching to be performed entirely in the specification space without additional data sharing.
>
> We thank the reviewer again for their valuable feedback, which will help us improve the clarity of the paper.

---

> > ### Author Rebuttal · Reviewer_B1jG · 2026-04-04
> >
> > Thanks for clarifying the compute cost of training $\phi_{\theta}$.
> >
> > For the question about comparing $U_c$ and $U_t$, I understand that they are both in R^d, but I meant specifically, that one is map from $\mathcal{H}_c \to \mathcal{G}$, and the other is a map from $\mathcal{H}_t \to \mathcal{G}$, where $\mathcal{H}_c$ and $\mathcal{H}_t$ are the RKHS of $\phi_c$ and $\phi_t$ respectively. Does it make sense to compare them like this when they expect different input objects?
> >
> > Regarding using $E_{x \sim P_X^t} \| U_c \phi(x) - U_t \phi(x) \|_G^2$, is this computation done client side? In that that this computation is done client-side? In which case wouldn't using $P_X^t$ be fine?

---

> > > ### Author Response · Authors · 2026-04-04
> > >
> > > We thank the reviewer for their insightful questions and careful follow-up. Our responses are below.
> > >
> > > ## On the comparability of $\widetilde{\mathcal{U}}_t$ and $\widetilde{\mathcal{U}}_c$
> > >
> > > The reviewer is correct that $\widetilde{\mathcal{U}}_t$ and $\widetilde{\mathcal{U}}_c$ have different domains: one maps from $\mathcal{H}_t$ to $\mathcal{G}$ and the other from $\mathcal{H}_c$ to $\mathcal{G}$. However, what we aim to compare is **not their outputs on specific inputs**, but rather the **input-output mapping relationship** they represent.
> > >
> > > Consider an analogy:
> > >
> > > - Let $f(x) = x^2$ be defined on $[-1, 1]$
> > > - Let $g(x) = (x-1)^2$ be defined on $[0, 2]$
> > >
> > > These two functions have different domains, yet we can still compare their **shapes** (i.e., the input-output relationship). Although $f(0)=0$ and $g(0)=1$ (the specific values differ), it does not prevent us from assessing whether their mapping structures are similar.
> > >
> > > Returning to our setting:
> > >
> > > - Both $\mathcal{H}_t$ and $\mathcal{H}_c$ are subspaces of $\mathbb{R}^d$, just as $[-1, 1]$ and $[0, 2]$ are subintervals of $\mathbb{R}$.
> > > - The Hilbert-Schmidt norm compares the **structure of the operator itself**. It depends only on the representation matrices of the operators with respect to a basis of the space.
> > >
> > > - The representation matrices of both operators are written under the standard basis of $\mathbb{R}^d$. Therefore, directly subtracting them and computing the HS norm still makes sense.
> > > - In addition, our experiments empirically validate that directly comparing $\widetilde{\mathcal{U}}_t$ and $\widetilde{\mathcal{U}}_c$ is effective.
> > >
> > > ## On the matching metric $\mathbb{E}\_{x\sim P_X^t}\left\\|\widetilde{\mathcal{U}}\_c\phi(x)-\widetilde{\mathcal{U}}\_t \phi(x)\right\\|_{\mathcal{G}}^2$ (eq1)
> > >
> > > In our original rebuttal, we mentioned that:
> > >
> > > > However, this would require the user to reveal their input distribution $P_X^t$, which may raise privacy concerns.
> > >
> > > The reviewer asks whether the computation of eq1 is done client-side. The answer is **no**, otherwise the market would need to **send all $\widetilde{\mathcal{U}}_c$ of all learnwares to the user**. This is impractical because existing model platforms are extremely large (e.g., Hugging Face hosts over 2.75M models). Even after pre-filtering a large number of irrelevant models using methods such as clustering, the user would still need to download and process hundreds or thousands of $\widetilde{\mathcal{U}}_c$ on the client side, which is both computationally expensive and time-consuming.
> > >
> > > Therefore, in the learnware paradigm, the user generates the privacy-preserving specification $\widetilde{\mathcal{U}}_t$ locally on the client side, and then uploads it to the learnware market. The market is responsible for matching and recommending the most suitable learnware, i.e., computing $c^*={\rm argmin}_c\left\\|\widetilde{\mathcal{U}}\_c-\widetilde{\mathcal{U}}\_t\right\\|\_{\rm HS}^2$. As we explained above, this direct comparison makes sense.
> > >
> > > Therefore, we believe that performing the matching on the market side and using the HS norm to directly compare $\widetilde{\mathcal{U}}_c$ and $\widetilde{\mathcal{U}}_t$ is more consistent with the design goals of the learnware paradigm.

---

### Official Review · Reviewer_euuJ · 2026-03-24

**Soundness:** 4
**Presentation:** 4
**Significance:** 2
**Originality:** 2
**Overall Recommendation:** 3
**Confidence:** 3

**Summary:**

This paper provides a methodology to identify learnwares (i.e., models on a public market) that takes into account both the inputs and outputs of the desired target task rather than only inputs, and is "privacy-preserving" in the sense that the specification of inputs and outputs does not involve raw samples. The evaluation shows that the proposed RNCME outperforms other methods in identifying appropriate models for each evaluated task.

**Compliance With Llm Reviewing Policy:**

Affirmed.

**Key Questions For Authors:**

I do not have specific questions that would change my recommendation.

**Limitations:**

As far as I can tell the authors have not discussed limitations in the work. One limitation to discuss is the privacy guarantees or lack thereof.

**Strengths And Weaknesses:**

Strengths:
- The paper proposes a thorough approach to solving the problem of identifying learnwares based on both inputs and outputs rather than only on inputs.
- The approach is clearly presented and is validated empirically on multiple datasets. Basic guarantees about the fidelity of the representation are backed by proofs.

Weaknesses:
- One weakness is that the paper claims that the reduction "preserves privacy" but it is unclear in what formal sense it preserves privacy. Privacy is a guarantee that should not be stated lightly and the set reduction does not seem to satisfy any known definition of privacy; at minimum, if the authors claim privacy preservation then this should be stated under an appropriate attacker threat model.
- A secondary weakness is that it is truthfully unclear to me, in spite of prior work on this topic, whether the learnware paradigm is actually likely to be relevant in practice. This is not a critique specific to this particular paper, but I am not fully convinced of the merit of accepting an incremental work in a field that does not appear to be grounded in a practically relevant problem.

---

> ### Author Rebuttal · Authors · 2026-03-26
>
> We thank the reviewers for their time and constructive feedback. Below we address the comments point by point.
>
> ### Weakness 1: On privacy preservation
>
> We appreciate the reviewer's concern. It is important to note that our RNCME specification does not directly expose raw data, and its privacy protection is not merely heuristic. Lei et al. [1] has rigorously analyzed the privacy property of RKME specification, demonstrating its resistance to linkage and inference attacks, and showing that effective privacy protection can be achieved solely through compression, even without introducing differential privacy. RNCME can be viewed as a vector-valued extension of RKME to the output space, and its specification similarly consists of compressed samples and weights. Therefore, RNCME naturally fits into the theoretical privacy analysis framework established in [1]. In fact, the neural coupling and explicit norm constraints in RNCME may further enhance privacy protection. We thank the reviewer for pointing out that the original text was not clear enough on this point, and we will explicitly supplement the above discussion in the revised manuscript.
>
> ### Weakness 2: On practical relevance
>
> We understand the reviewer's concern regarding the learnware paradigm. However, regardless of the term "learnware", characterizing, identifying, and reusing massive trained models has become an important trend in machine learning. Recent works on model embedding, model zoo retrieval, and model capability representation all address the same fundamental question: how to use a "specification" to describe model functionality for efficient and safe reuse. This technical route was first clearly proposed by the learnware paradigm and has been validated in real systems. For instance, the first learnware dock system, Beimingwu [2, 3], has been open-sourced and deployed online, with thousands of researchers from over 150 universities registered. Moreover, relevant technologies have been incubated through industry collaborations. Thus, the specification-based approach has broad practical prospects beyond a specific academic concept. We will add this discussion to the paper to help readers better appreciate the practical foundation of the specification route.
>
> ### References
>
> 1. Lei, H.-Y., Tan, Z.-H., and Zhou, Z.-H. On the ability of developers’ training data preservation of learnware. In Advances in Neural Information Processing Systems, volume 37, pp. 36471–36513, 2024.
> 2. Tan, Z.-H., Liu, J.-D., Bi, X.-D., Tan, P., Zheng, Q.-C., Liu, H.-T., Xie, Y., Zou, X.-C., Yu, Y., and Zhou, Z.-H. Beimingwu: A learnware dock system. In Proceedings of the 30th ACM SIGKDD Conference on Knowledge Discovery, pp. 5773–5782. ACM, 2024b.
> 3. Website: https://bmwu.cloud/

---

> > ### Author Rebuttal · Reviewer_euuJ · 2026-04-03
> >
> > Thank you for your response. I appreciate the authors' pointing me to the prior work on privacy of this paradigm. While I made a best-effort to read and understand that prior work, the privacy methodology proposed there is new in the literature and I don't have the ability to meaningfully evaluate it or contextualize it in the more prominent privacy methodologies that I am familiar with (e.g., differential privacy). I read through the reviews of that prior work and found that those reviewers also focused primarily on empirical considerations and did not thoroughly evaluate the theoretical claims put forth, but evaluating the prior work is out of scope for my job as a reviewer of the current paper.
> >
> > I would prefer to keep my score, but if other reviewers are more familiar with the prior privacy methodology that the authors here rely on, I will defer to those reviewers.

---

> > > ### Author Response · Authors · 2026-04-04
> > >
> > > We thank the reviewer for their honest feedback and for making the effort to read the prior work by Lei et al. (2024). We fully respect that the reviewer is not familiar with this privacy methodology and therefore prefers to keep their score while deferring to other reviewers.
> > >
> > > We would like to reiterate that Lei et al. (2024) formally proved that reduced sets preserve training data privacy, even under model inversion attacks. Our RNCME adopts a similar reduced set to protect privacy. Furthermore, the neural feature map $\phi\_\theta$ and the matrix inversion step in CME estimation provide additional layers of protection by extracting only task-relevant representations without exposing raw inputs.
> > >
> > > That said, the primary contribution of our work is to propose RNCME as a **specification generation method** that captures input-output conditional distributions and extends the learnware paradigm to both regression and classification tasks. A rigorous formal proof of privacy guarantees is not the main focus of this paper. We will conduct a thorough privacy analysis under a well-defined threat model as part of our future work. In the revised manuscript, we will also make the connection to Lei et al. (2024) much clearer.
> > >
> > > We appreciate the reviewer's time and understanding.
> > >
> > >
> > >
> > > ## Reference
> > >
> > > Lei, H.-Y., Tan, Z.-H., and Zhou, Z.-H. On the ability of developers’ training data preservation of learnware. In Advances in Neural Information Processing Systems, volume 37, pp. 36471–36513, 2024.

---

### Decision · Program_Chairs · 2026-04-30

**Decision:**

Accept (regular)

**Comment:**

The paper presents a conditional kernel mean embedding framework for learnware tasks. Two of the reviewers showed enthusiasm about the paper, mentioning (S1) a theoretical bound which lacked in prior work; and (S2) strong empirical evidence. However, two other reviewers showed hesitation, mentioning (W1) that the privacy guarantee in the reduced set isn't formal, like in differential privacy; (W2) some missing references that the authors mentioned but the reviewer could not find, and (W3) the debate over whether the Tan et al paper does regression or not.

About the point (W1), I agree. As someone working on more formal guarantees than on a method being robust against certain attack scenarios, I also find papers that bluntly claim their method is privacy-preserving not rigorous. They should clearly mention in what sense the method is so. Clearly, the reduced set doesn't have such a formal guarantee as in differential privacy.

Regarding point (W2), I read the authors' confidential message to AC, but I think the reviewer is referring to Figure 6 in Appendix E.3. In fact, the submitted paper contains three figures, and I couldn't find Fig 6 either. I wonder whether the authors are looking at their own version rather than the one they submitted to ICML.

Despite these weaknesses, I think the strengths (S1) and (S2) outweigh them, and the weaknesses do not undermine the soundness and integrity of their work and can be addressed straightforwardly. Hence, I suggest acceptance of the paper.